



# State updating in the Xin'anjiang Model: Joint assimilating streamflow and multi-source soil moisture data via Asynchronous Ensemble Kalman Filter with enhanced Error Models

Junfu Gong[1,2], Xingwen Liu[2], Cheng Yao[2,3]*, Zhijia Li[2], Albrecht H. Weerts[4,5], Qiaoling Li[2], Satish
Bastola[6], Yingchun Huang[7], Junzeng Xu[1]

[1]College of Agricultural Science and Engineering, Hohai University, Nanjing, 210024, China
[2]College of Hydrology and Water Resources, Hohai University, Nanjing, 210024, China
[3]China Meteorological Administration Hydro-Meteorology Key Laboratory, Nanjing, 210024, China
[4]Deltares, Delft, 2600 MH, Netherlands
[5]Hydrology and Environmental Hydraulics Group, Wageningen University, Wageningen, 6700 HB, Netherlands
[6]Department of Civil and Environmental Engineering, University of New Orleans, New Orleans, 70148, USA
[7]College of Civil Engineering, Fuzhou University, Fuzhou, 350108, China

*Correspondence to*: Cheng Yao (yaocheng@hhu.edu.cn)

**Abstract.** Assimilating either soil moisture or streamflow individually has been well demonstrated to enhance the simulation
performance of hydrological models. However, the runoff routing process may introduce a lag between soil moisture and
outlet discharge, presenting challenges in simultaneously assimilating the two types of observations into a hydrological
model. The Asynchronous Ensemble Kalman Filter (AEnKF), an adaptation of the Ensemble Kalman Filter (EnKF), is
capable of utilizing observations from both the assimilation moment and preceding periods, thus holding potential to address
this challenge. Our study first merges soil moisture data collected from field soil moisture monitoring sites with China
Meteorological Administration Land Data Assimilation System (CLDAS) soil moisture data. We then employ the AEnKF,
equipped with improved error models, to assimilate both observed outlet discharge and the merged soil moisture data into
the Xin'anjiang model. This process updates the state variables of the model, aiming to enhance real-time flood forecasting
performance. The testing on both synthetic and real-world cases demonstrates that assimilation of these two types of
observations simultaneously substantially reduces the accumulation of past errors in the initial conditions at the start of the
forecast, thereby aiding in elevating the accuracy of flood forecasting. Moreover, the AEnKF with the enhanced error model
consistently yields greater forecasting accuracy across various lead times compared to the standard EnKF.

## 1 Introduction

Floods, among the most frequent natural disasters, significantly affect infrastructure and agricultural yields and may even
directly endanger the lives of local residents (Johnson et al., 2020). The destructiveness of flash floods is particularly notable.
In recent decades, flash floods triggered by localized torrential rains have frequently resulted in significant human casualties
(Pilon, 2002). Short-term flood forecasting, a vital non-structural approach to flood mitigation, plays a crucial role in





facilitating emergency responses in flood-prone regions (Craninx et al., 2021). Hydrological models are instrumental in flood forecasting, utilizing mathematical and physical representations to analyze the various components of the catchment hydrological processes, including precipitation, evaporation, and runoff, as well as their interplay. This understanding aids in comprehending catchment hydrological characteristics and trends, crucial for simulating and forecasting hydrological processes. The Xin'anjiang model, extensively applied in operational short-term flood forecasting in China, stands as one of the well-known semi-distributed hydrological models. Its broad applicability, especially in the humid and semi-humid climate zones of the Yangtze River Basin, has been substantiated by extensive studies (e.g. Fang et al., 2017; Gong et al., 2021; Zang et al., 2021).

However, in hydrological simulations, multiple sources of uncertainty, such as uncertainties in model inputs, structure, and parameters, can significantly affect the accuracy of the simulations (Beven, 1993; Ajami et al., 2007). In short-term flood forecasting, an additional process, often referred to as the real-time correction process, is typically employed to mitigate these uncertainties. A notable strategy in real-time correction involves the recursive adjustment of the hydrological model's state variables based on available real-time observational data. It helps reduce the error accumulation in the initial conditions of hydrological model, a factor that has been identified as a primary source of uncertainty at the start of flood forecasting (Shukla and Lettenmaier, 2011; Yossef et al., 2013; Thiboult et al., 2016). This process is sometimes termed hydrological data assimilation in literature (e.g. Clark et al., 2008). The Ensemble Kalman Filter (EnKF) (Evensen, 2003), which integrates ensemble forecasting concepts with Kalman filter and employs Monte Carlo methods for error statistic prediction, effectively addresses the inability of Kalman filtering to handle nonlinear systems. Its robustness, flexibility, and ease of use have led to its widespread application in hydrological data assimilation (Clark et al., 2008; Liu et al., 2012; Rakovec et al., 2012; Piazzi et al., 2021).

Data assimilation typically falls into two categories: synchronous and asynchronous methods. Synchronous methods depend solely on observational data at a specific update moment, while asynchronous methods broaden this scope by incorporating data over a time frame, including both current and preceding time steps (Sakov and Bocquet, 2018). This distinction is particularly crucial in sequential assimilation, where commonly employed sequential filters like the EnKF utilize a synchronous strategy. Conversely, the asynchronous strategy is predominantly used in smoothers, such as the Ensemble Kalman Smoother (EnKS) (Evensen and Van Leeuwen, 2000). While the EnKS augments reanalysis by integrating future observational data backwards in time, its forecasting efficacy (including real-time forecasting) aligns with that of the EnKF (Evensen, 2009). The intrinsic difference between smoothers and filters is their focus: smoothers assimilate future observational data, while filters process past observational data (Rakovec et al., 2015). Hence, in hydrological data assimilation with a focus on forecasting, filters are generally the preferred choice over smoothers.

In recent years, researchers have made strides in integrating asynchronous strategies into filters for sequential assimilation. This is notably evident in the development of the Four-Dimensional Ensemble Kalman Filter (4D-EnKF) (Hunt et al., 2004) and the Four-Dimensional Local Ensemble Transform Kalman Filter (4D-LETKF) (Hunt et al., 2007). The 4D-EnKF stands out for its ability to synchronize the timing of observations with lower computational demands, particularly effective in





linear dynamics. In contrast, the 4D-LETKF builds upon the 4D-EnKF by prioritizing spatial localization and refining the handling of nonlinear observation operators. This enhancement renders it more effective and versatile in managing high-dimensional, chaotic systems, especially in meteorology and climatology. Building on this, Sakov et al. (2010, 2018) introduced the asynchronous ensemble Kalman filter (AEnKF). Remarkably, the AEnKF and 4D-LETKF are essentially

equivalent (Sakov et al., 2010), both employing ensemble-based methods to update model states based on observational data. The 4D-LETKF processes asynchronous observations by amalgamating them and updating the state via ensemble transform matrices. Conversely, the AEnKF accomplishes this by advancing corrections along the forecast system trajectory, utilizing ensemble observations from the observation time, thereby efficiently assimilating both past and future data. AEnKF is designed to be computationally efficient, which is noted for its relative simplicity in implementation compared to 4D-

LETKF. It modifies the standard EnKF by using ensemble observations from the time of observations, a straightforward change that does not significantly complicate the assimilation process. The AEnKF is recognized for its simplicity and high computational efficiency, offering significant potential in short-term flood forecasting applications. Despite its promise, the scope of research in this area is relatively limited. Among the few studies conducted, Mazzoleni et al. (2018) evaluated AEnKF assimilation in simplified flow routing models, highlighting its exceptional performance in both lumped and

distributed flow routing. In addition, Rakovec et al. (2015) and our earlier study (Gong et al., 2024) applied the AEnKF to the distributed HBV-96 model and the Xin'anjiang model, respectively. These studies examined effectiveness of AEnKF in real-time correction through the assimilation of observed discharge in distributed and semi-distributed hydrological models, revealing that AEnKF outperforms the standard EnKF. However, these studies assimilate only a single type of observational data (e.g., observed discharge) using the AEnKF method, which does not take full advantage of the AEnKF.

In the context of real-time correction processes employing AEnKF, the types of observations to assimilate constitute another key factor influencing the effectiveness. Popular observation types currently assimilated include discharge, soil moisture, and snow data, among others (Gong et al., 2023). In rainfall-runoff modeling, soil moisture plays a pivotal role in driving the runoff generation process (Massari et al., 2014). A wealth of research has demonstrated that updating hydrological model states through the assimilation of soil moisture significantly enhances the precision of runoff simulations and forecasts (e.g.,

Wanders et al., 2014; Alvarez-Garreton et al. 2015; Chao et al., 2022). These studies typically rely on a single type of soil moisture dataset. One of the highlights of our study is that it simultaneously considers the advantages of site observation data and soil reanalysis datasets, enhancing both the timeliness and spatial accuracy of the soil moisture data. Specifically, in the real-time correction process of flood forecasting, there is a high demand for the timeliness of observational data to swiftly respond to flood events. Satellite remote sensing data and reanalysis products often suffer from delays in data release or

lengthy observational intervals. In contrast, ground-based soil moisture measurements offer high accuracy and timeliness but are limited to point-scale data, failing to capture the spatial distribution of soil moisture. To overcome this limitation, the Weighted k-Nearest Neighbor (WKNN) algorithm (Pedregosa et al., 2011; Jung et al., 2017) is employed to fuse ground soil moisture measurements with reanalysis soil moisture data. This approach involves establishing a regression relationship between historical ground and reanalysis data, subsequently generating real-time, spatially distributed, fusion soil moisture



data from current ground observations. On the other hand, discharge observations, due to their direct relevance to flow or water level predictions crucial in flood forecasting, are another valuable choice for assimilation. They provide a comprehensive view of the hydrological conditions of a catchment. Discharge measurements are often more accessible and offer more timely data than soil moisture readings, generally yielding greater reliability (Li et al., 2013). Numerous studies have concentrated on assimilating observed discharge data to enhance flood forecasting, showcasing the substantial potential
and impressive effectiveness of this strategy across various regions (e.g., Clark et al., 2008; Sun et al., 2020; Gong et al., 2023). Given that assimilating soil moisture or discharge alone can provide acceptable results, exploring the simultaneous assimilation of both observation types warrants consideration. Previous studies have highlighted the benefits of concurrently assimilating various observation types. Techniques such as the EnKF (Meng et al., 2017), Variational Assimilation (VAR) (Lee et al., 2011), and Tempered Particle Filter (TPF) (García-Alén et al., 2023) have consistently shown that joint
assimilation generally surpasses the efficiency of single-type assimilation. Although these findings are encouraging, the advantage of joint assimilation may not always hold. This is partly because each observation type represents a specific hydrological process, with correlations among variables varying across different spatial and temporal scales. For instance, soil moisture immediately responds to rainfall, while streamflow responses are inherently delayed due to the time delay in the routing process (Meng et al., 2017). Such delays can lead to the accumulation of uncertainties in discharge predictions,
an aspect often overlooked in synchronized assimilation methods. Contrarily, the AEnKF method considers all observational data within a specific time window, rather than just a single observation at the update time, effectively considering the time delays in routing processes and offering a novel approach for the combined assimilation of diverse observation types. However, to our knowledge, there are no existing studies on the performance of AEnKF in assimilating multiple types of observational datasets (such as soil moisture and discharge measurements), which could significantly improve the accuracy
of short-term flood forecasting.

In AEnKF assimilation, ensemble dispersion is achieved by introducing pre-determined noise (commonly zero-mean Gaussian noise) into model state variables and forcing data. The models governing these perturbations are termed 'error models', and their associated parameters are known as 'hyperparameters' (Thiboult and Anctil, 2015). Improper handling of these uncertainties can potentially impair the efficacy of ensemble-based Kalman filters (Crow and Van Loon, 2006;
Pathiraja et al., 2018). The commonly adopted practice involves setting the hyperparameters of error models based on the empirical knowledge of hydrologists or forecasters (e.g., Weerts and El Serafy, 2006; Clark et al., 2008; Sun et al., 2020). This approach is highly subjective, resulting in forecast results that may significantly differ among practitioners. The Maximum a posteriori estimation (MAP) method (Li et al., 2014; Gong et al., 2023) represents a Bayesian inference technique specifically designed for ensemble-based Kalman filters. This method leverages historical observational data to
objectively estimate the hyperparameters in error models, thereby substantially mitigating the subjectivity associated with hyperparameter configuration. Notably, the strengths of MAP method, compared to alternatives like the kernel conditional density estimation method (Pathiraja et al., 2018), include its independence from the need for sequential observations to be independent. Furthermore, it enables concurrent estimation of hyperparameters across diverse error models, making it





particularly compatible with the error models employed in AEnKF. Another challenge in AEnKF assimilation is reducing
the systematic biases that arise from perturbations. When creating ensemble dispersion using error models, it is implicitly
assumed that the introduction of noise will not lead to systematic biases in the model outputs (Ryu et al., 2009). Nevertheless,
the strong non-linearity of hydrological models and the stringent physical limitations on some state variables mean that even
zero-mean Gaussian perturbations may result in systematic biases (Alvarez-Garreton et al., 2015). A case in point is soil
moisture, which must stay below saturation levels. During flooding, when soil moisture approaches saturation, perturbing
this variable risk breaching these physical boundaries. Subsequent corrections made by the hydrological model to align with
saturation levels may introduce truncation errors in the prediction of the background field. To counter this, our study
incorporates the Bias-corrected Gaussian Error Model (BGEM) (Ryu et al., 2009), which introduces an unperturbed model
run in parallel to the ensemble. This unperturbed model is utilized to correct the biases induced by perturbations. Our prior
research (Gong et al., 2024) has shown that the BGEM is effective in alleviating systematic biases caused by random
perturbations in soil moisture state variables. However, the performance of the AEnKF with these enhanced error models
when assimilating multiple types of observations has yet to be further tested.

This study developed an efficient joint data assimilation framework for real-time correction of short-term flood forecasting
based on AEnKF with improved error models. One of the main highlights of this study is the consideration of the inherent
limitations of single-source soil moisture data. By fusing ground-based soil moisture measurements with reanalysis data
from the China Meteorological Administration Land Data Assimilation System (CLDAS), the study generates a reliable,
real-time spatial distribution dataset of soil moisture that aligns with the 8-hour observation intervals of monitoring sites. The
second highlight is that the AEnKF with improved error models fully accounts for the time delays in routing process,
enabling effective joint assimilation of soil moisture data and discharge observations. Upon establishing the appropriate
assimilation time window for the AEnKF with improved error models, the study conducted a detailed comparison between
the joint assimilation scheme and individual assimilation schemes (including the separate assimilation of soil moisture or
discharge observation data) using synthetic and real-world cases. This comparison effectively underscores the superior
performance of the joint assimilation framework proposed in this study.

## 2 Methodology and method

### 2.1 Asynchronous Ensemble Kalman filter

The Asynchronous Ensemble Kalman Filter (AEnKF) represents a straightforward enhancement of the Ensemble Kalman
Filter (EnKF), utilizing the same assimilation framework as EnKF. Its uniqueness lies in its capability to assimilate multi-
temporal observational data, enabling it to effectively incorporate a broader temporal spectrum of observations. This feature
is particularly advantageous in capturing the dynamic nature of hydrological processes over time. We follow the notation of
Ide et al. (1997) and Vetra-Carvalho et al. (2018) as closely as possible, aiming to make our paper accessible and practical
for both data assimilation specialists and a broader audience interested in applying these methods. To this end, the dimension





of the state space, observation space is denoted as $N_x$ and $N_y$. Further, the time index is always denoted in parentheses to the right of the variable, i.e. $(.)(t_i)$. Notably, the observational data are categorized into two types: the observed discharge at the catchment outlet and soil moisture across sub-basins. During an ensemble run of the dynamic model, the assimilation process has two steps: the soil moisture observations are used to update the soil states, and the discharge observations are

used to update cumulative channel flow. Consequently, for each assimilation process, the values of $N_y$ can differ, and the same applies to $N_x$.

### 2.1.1 Ensemble Kalman filter

At a given time $t_i$, we define the model state vector as $\boldsymbol{x}(t_i) \in \mathcal{R}^{N_x}$ and the observation vector as $\boldsymbol{y}(t_i) \in \mathcal{R}^{N_y}$. In the EnKF framework, it is crucial to generate a set of independent model state vectors. These vectors constitute an ensemble matrix,

denoted as $\boldsymbol{X}(t_i) \in \mathcal{R}^{N_x \times N_e}$, where $N_e$ is the total number of the ensemble members. The initial $\boldsymbol{X}(0)$ is obtained by the Monte Carlo method.

The state transfer equation at the forecast step is represented by

$$\boldsymbol{x}_j^f(t_{i+1}) = \mathcal{M}[\boldsymbol{x}_j^a(t_i), \boldsymbol{U}(t_i)] + \boldsymbol{\eta}(t_i) \tag{1}$$

Where $\mathcal{M}[.]: \mathcal{R}^{N_x} \to \mathcal{R}^{N_x}$ signifies the dynamic model, such as the Xin'anjiang model; $\boldsymbol{U}(t_i)$ represents the forcing data (including rainfall, evaporation, etc.); $\boldsymbol{\eta}(t_i) \in \mathcal{R}^{N_x}$ symbolizes the process or system noise characterized by a mean of zero

and a covariance matrix $\boldsymbol{Q}(t_i)$. In addition, the subscript '$j$' signifies the ensemble index, ranging from 1 to $N_e$. The forecasted values from the dynamic model are marked with a superscript '$f$', while the analysis (updated) values from the filter are denoted by a superscript '$a$'.

During the analysis step, we create a set of new observation vectors by perturbing the original observation vector $\boldsymbol{y}(t_i)$, as described by

$$\boldsymbol{y}_j^o(t_i) = \boldsymbol{y}(t_i) + \boldsymbol{\varepsilon}(t_i) \tag{2}$$

Where $\boldsymbol{y}_j^o(t_i) \in \mathcal{R}^{N_y}$ represents the perturbed observation vector for the $j^{\text{th}}$ ensemble, and $\boldsymbol{\varepsilon}(t_i) \in \mathcal{R}^{N_y}$ is Gaussian noise characterized by covariance matrix $\boldsymbol{R}(t_i)$. We assume spatial independence of observation errors, thereby designating $\boldsymbol{R}(t_i)$ as a diagonal matrix. Furthermore, the state update equation is expressed as follows:

$$\boldsymbol{x}_j^a(t_i) = \boldsymbol{x}_j^f(t_i) + \boldsymbol{K}(t_i) \cdot (\boldsymbol{y}_j^o(t_i) - \mathcal{H}[\boldsymbol{x}_j^f(t_i)]) \tag{3}$$

Where $\mathcal{H}[.]$ is the measurement operator that maps the state space to observation space, which is also Xin'anjiang model in this study, and $\boldsymbol{K}(t_i)$ is the Kalman gain matrix calculated by the following:

$$\boldsymbol{K}(t_i) = \boldsymbol{P}^f(t_i)\boldsymbol{H}^T[\boldsymbol{H}\boldsymbol{P}^f(t_i)\boldsymbol{H}^T + \boldsymbol{R}(t_i)]^{-1} \tag{4}$$

In scenarios where the state space dimensionality, $N_x$, is substantial, bypassing the direct computation of $\boldsymbol{P}^f(t_i)$ in favor of calculating $\boldsymbol{P}^f(t_i)\boldsymbol{H}^T$ and $\boldsymbol{H}\boldsymbol{P}^f(t_i)\boldsymbol{H}^T$ emerges as a strategy to enhance computational efficiency, as highlighted by Nerger and Hiller (2013).



### 2.1.2 Asynchronous variant

AEnKF is based on the concept of joint state-observation space, where the ensemble is replaced by a joint ensemble that

combines state and observation information. Updating model states involves considering observations from both the current and previous time steps, controlled by the assimilation time window, $tw$. This window defines the duration over which observations are considered for the analysis, for instance, including data from the previous five hours. Moreover, when assimilating only current observations ($tw = 0$), the AEnKF reverts to standard EnKF. In the AEnKF, the observation vector is altered to:

$$\widetilde{\boldsymbol{y}}(t_i) = [\boldsymbol{y}(t_i)^T, \boldsymbol{y}(t_{i-1})^T, \ldots, \boldsymbol{y}(t_{i-tw})^T]^T \in \mathcal{R}^{(tw+1)*N_y} \tag{5}$$

Where $\widetilde{\boldsymbol{y}}(t_i)$ is the joint observation vector, and $\widetilde{\boldsymbol{R}}(t_i)$ denotes the covariance matrix of the associated observation noise, expressed as a diagonal matrix:

$$\widetilde{\boldsymbol{R}}(t_i) = \begin{bmatrix} \boldsymbol{R}(t_i) & \cdots & & 0 \\ \vdots & \boldsymbol{R}(t_{i-1}) & & \vdots \\ & & \ddots & \\ 0 & \cdots & & \boldsymbol{R}(t_{i-tw}) \end{bmatrix} \tag{6}$$

Similarly, the model prediction vector from the prior $tw$ time steps in the observation space is used to expand the state vector:

$$\widetilde{\boldsymbol{x}}_j^f(t_i) = \left(\boldsymbol{x}_j^f(t_{i+1})^T, \mathcal{H}[\boldsymbol{x}_j^f(t_{i-1})]^T, \mathcal{H}[\boldsymbol{x}_j^f(t_{i-2})]^T, \ldots, \mathcal{H}[\boldsymbol{x}_j^f(t_{i-tw})]^T\right)^T \in \mathcal{R}^{N_x+tw*N_y} \tag{7}$$

Furthermore, the new state definition introduces an augmented observation operator $\widetilde{\mathcal{H}}(t_i)$:

$$\widetilde{\mathcal{H}} = \begin{bmatrix} \mathcal{H} & \cdots & & 0 \\ \vdots & I_i & & \vdots \\ & & \ddots & \\ 0 & \cdots & & I_{i-tw} \end{bmatrix} \tag{8}$$

Where $I$, with the corresponding subscript, stands for identity elements on the diagonal, matching the dimensions in Eq. (7).

Following these augmented equations for $\widetilde{\boldsymbol{x}}_j^f(t_i)$, $\widetilde{\boldsymbol{y}}(t_i)$, $\widetilde{\boldsymbol{R}}(t_i)$, and $\widetilde{\mathcal{H}}$, we can directly apply these augmented variables in the EnKF process (Section 2.1.1) to implement the AEnKF assimilation. Crucially, in the joint state vector $\widetilde{\boldsymbol{x}}_j^f(t_i)$, model prediction vectors within the observation space, such as $\mathcal{H}[\boldsymbol{x}_j^f(t_{i-1})]$ and others, are considered diagnostic variables instead of state variables. As a result, they are not updated during the analysis step. Specifically, in Eq. (3), only the first $N_x$ elements of the vector $\widetilde{\boldsymbol{x}}_{n,t}$ are calculated, while others are disregarded.

### 2.2 Error estimation


Both the EnKF and its variant, update model states by employing a weighted average of observational data and model forecasts. This process highlights the crucial role of model and observational errors in determining the effectiveness of the assimilation system. Particularly in rainfall-runoff modeling, where uncertainties in both model and observations are inherently ambiguous, generalizing these uncertainties is instrumental in acquiring refined approximations of suboptimal

model states. A common technique involves adding unbiased noise to observations, model forcing and model states.





Observations involved in this study include discharge at catchment outlet and observed soil moisture. We generalize the observational errors as Gaussian perturbations related to the corresponding observed values (Weerts and El Serafy, 2006; Clark et al., 2008; Alvarez-Garreton et al., 2015). Given that rainfall serves as the most critical input information for the hydrological model, we employ log-normal multiplicative perturbations to describe the errors associated with rainfall,
thereby representing the uncertainty in model forcing (McMillan et al., 2011; DeChant and Moradkhani, 2012). Moreover, we introduce a first-order autoregressive model to represent the temporal correlation within the observational errors and the forcing errors.

In the assimilation of observed discharge at catchment outlet, the key model state variable updated is cumulative channel flow. This variable represents the outflow from each sub-basin on the routing calculation unit (sub-reaches in this study),
denoted as $QC$. As Li et al. (2014), this state variables are perturbed using a Gaussian function. When assimilating observed soil moisture, the model state variables representing soil humidity need to be updated. Specifically, this refers to the tension water storage (including upper, and lower layer tension water) and the free water storage in the Xin'anjiang model. In the Xin'anjiang model, the soil moisture state variables receive physical constraints. The free water storage (denoted as $S$) reflects the soil moisture in the topsoil layer, specifically the humus layer (Yao et al., 2012). Therefore, it is assumed that the
free water storage can be considered to range between the saturation moisture content and the field capacity, with its upper limit controlled by the parameter $SM$ and the lower limit set to zero. On the other hand, the tension water storage (denoted as $W$) represents the soil moisture throughout the entire soil profile, encompassing the whole unsaturated zone (Yao et al., 2012). Consequently, the tension water storage is considered to vary between the field capacity and the wilting point, with its upper limit governed by the parameter $WM$ and the lower limit being zero. The $WU$, $WL$, and $WD$ represent the upper, lower,
and deep layer tension water storage, respectively, with their upper limits controlled by the parameters $WUM$, $WLM$, and $WDM$, and $WM = WUM + WLM + WDM$. When the variables approach the upper or lower limit, the Gaussian perturbations may cause it to violate the physical constraints. If the hydrological model corrects it, it will lead to the truncation error in the background field predictions. We introduce the Bias-corrected Gaussian Error Model (BGEM) proposed by Ryu et al. (2009), aimed at reducing biases that emerge due to adherence to physical constraints.
The aforementioned error models are controlled by parameters known as 'hyperparameters' (Thiboult and Anctil, 2015), such as the hyperparameters for Gaussian perturbations are mean and standard deviation. We apply the Maximum a posteriori estimation method (MAP) to identify the globally optimal values of these hyperparameters (Gong et al., 2023). The MAP method aims to maximize the probability density of the hyperparameters with given the observed historical flood events. Supplement 1 provides a comprehensive introduction to the implementation of error estimation in this study.

**2.3 Hydrological model**

The Xin'anjiang model, conceptualized by Zhao (1992), is a distinguished hydrological model, primarily based on a saturation excess mechanism. Renowned for its straightforward structure and explicit parameter definitions, this model excels in simulating humid catchments, making it a popular tool for flood forecasting in in China. To account for spatial





variability in rainfall distribution and surface characteristics, the model typically segments a catchment into several sub-
basins. These sub-basins act as computational units for runoff generation and routing.

The Xin'anjiang model demands relatively straightforward driving data, and key inputs include the areal mean rainfall depth (P) and pan evaporation (EM) for each sub-basin. The model typically comprises four main components: evapotranspiration, runoff production, runoff separation, and flow routing, involving the calibration of 16 distinct parameters. The flow chart of the Xin'anjiang model is presented in Fig. 1. Soil evaporation is derived from pan evaporation data using a 'three-layer soil
moisture module'. The runoff generation is based on a saturation-excess mechanism, where runoff is produced only when the soil moisture in the unsaturated zone reaches field capacity. The 'lag and route' method calculates the outflow from each sub-basin. Flow routing from the sub-basin outlets to the total basin outlet employs the Muskingum method to successive sub-reaches. It is implemented through dividing the channel from each sub-basin outlet to the total basin outlet into varying numbers of sub-reaches. These sub-reaches are based on the distance from each sub-basin outlet to the total basin outlet. In
addition, the basin inflow is directly calculated to the outlet by the Muskingum method.

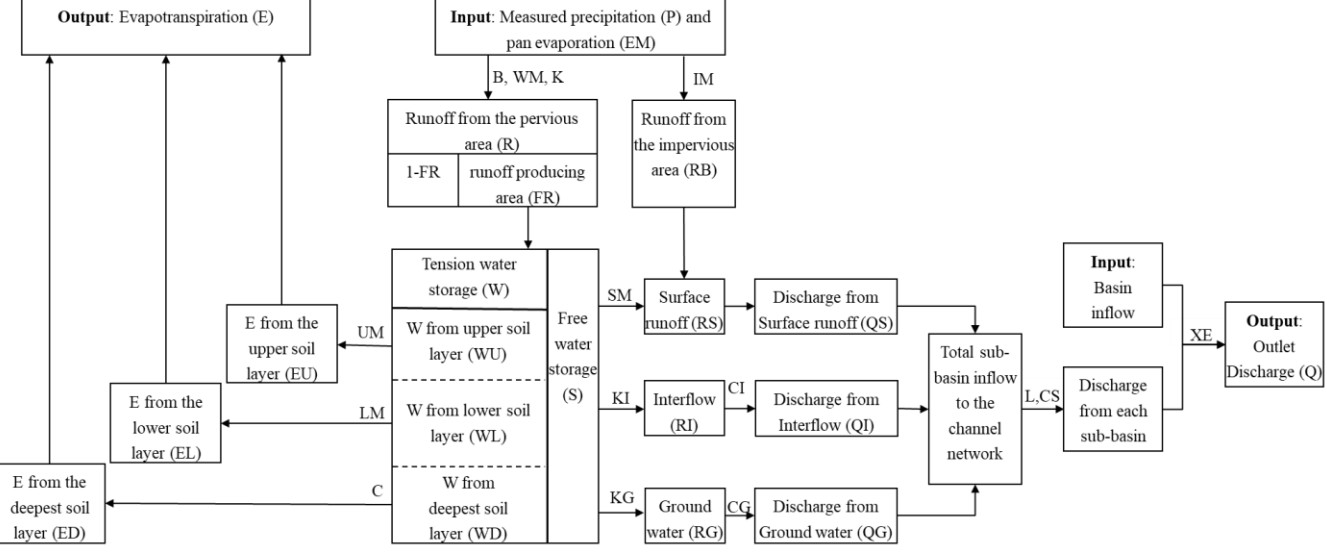

**Fig. 1. Flow Chart of Xin'anjiang Model. The variables in the boxes indicate the model state, inputs and outputs, and the symbols outside the corresponding blocks are model parameters.**

Zhao (1992) categorized the parameters of Xin'anjiang model into sensitive and non-sensitive groups. In real-world cases,
non-sensitive parameters are assigned values based on expert judgment, while optimal values for sensitive parameters are derived from historical data using the Shuffled Complex Evolution (SCE-UA) method (Duan et al., 1992). For synthetic cases, however, parameters are taken as recommend defaults. Table 1 summarizes these parameters.

**Table 1. Parameters of the Xin'anjiang model**

| Parameter [a] | Description | Synthetic cases | Real-world cases |
|---|---|---|---|
| ***K*** | the ratio of potential evapotranspiration to pan evaporation | 1.00 | 0.95 |





| Parameter [a] | Description | Synthetic cases | Real-world cases |
|---|---|---|---|
| C | Evapotranspiration coefficient of deeper layer | 0.13 | 0.05 |
| WUM | Averaged tension water capacity of upper layer (mm) | 12.5 | 19.9 |
| WLM | Averaged tension water capacity of lower layer (mm) | 75.0 | 64.4 |
| B | Exponent of the tension water capacity curve | 0.40 | 0.38 |
| WM | Averaged tension water capacity (mm) | 125.0 | 119.8 |
| IM | Percentage of impervious areas in the catchment | 0.01 | 0.03 |
| **_SM_** | Averaged free water storage capacity (mm) | 30.0 | 16.7 |
| EX | Exponent of the free water capacity curve | 1.25 | 1.50 |
| **_KI_** | Daily outflow coefficient of free water storage to interflow | 0.35 | 0.02 |
| **_KG_** | Daily outflow coefficient of free water storage to groundwater | 0.35 | 0.68 |
| CI | Daily recession constant of the interflow storage | 0.70 | 0.52 |
| **_CG_** | Daily recession constant of the groundwater storage | 0.99 | 0.93 |
| **_CS_** | Daily recession constants of channel network storage | 0.50 | 0.88 |
| **_LAG_** | Lag in time (h) | 0 | 1 |
| XE | Parameters of the Muskingum method | 0.25 | 0.01 |

[a] Parameters in bold and underline text indicate sensitive parameters.

## 2.4 Multi-source soil moisture data fusion

The soil moisture reanalysis data are sourced from the China Meteorological Administration Land Data Assimilation System (CLDAS) near-real-time dataset (https://data.cma.cn/). While the CLDAS dataset demonstrates a reasonable level of accuracy within China, with a regional average correlation coefficient of 0.89, a root mean square error of 0.02 m³/m³, and a bias of 0.01 m³/m³ (Wang and Li, 2020), it faces limitations due to missing values in some areas and data latency (published with a two-day lag), restricting its application in real-time flood forecasting in small and medium-sized catchments. On the other hand, ground station measurements offer high precision and timeliness (real-time data) but represent point-scale soil moisture, while the Xin'anjiang model simulates soil moisture as areal averages for sub-basin, necessitating consideration of spatial scale effects. To bridge this gap and assimilate soil moisture observations into the Xin'anjiang model, this study employs weighted *k* nearest neighbor (WKNN) algorithm (Pedregosa et al., 2011) to merge CLDAS soil moisture data (hereinafter referred to as CLDAS) with in-suit soil moisture data collected from monitoring sites (hereinafter referred to as IN-SUIT). This method generates real-time, spatially distributed soil moisture data, based on in-situ observations and the spatial distribution from the CLDAS dataset, ensuring compatibility with the tension water storage and free water storage in the Xin'anjiang model.

The Harmonized World Soil Database (HWSD) provides soil texture map for two layers: 0-30 cm (topsoil layer, T) and 30-100 cm (subsoil layer, S). Initially, using the technique by Reynolds et al. (2000), soil transfer functions (PTFs) are applied to the grid of soil texture map. This process involves estimating wilting point $\theta_{wp}$, field capacity $\theta_{fc}$, and saturation moisture





content $\theta_s$ for each grid layer based on its soil clay and sand percentage contents, along with USDA soil texture classification. In this study, we assume that the soil moisture constants for each sub-basin are the arithmetic average of the grid-scale soil moisture constants within the corresponding areas.

In the Xin'anjiang model, tension water capacity (*WM*), corresponding to available water capacity, is defined as the moisture content between the wilting point and field capacity, thus representing the thickness of the unsaturated zone. Free water capacity (*SM*) is defined as the moisture content between field capacity and saturation moisture content, relating to the thickness of the humus soil layer. Accordingly, we define a conceptual soil profile in the Xin'anjiang model, where the soil profile of tension water is divided into upper, lower, and deep layers. The capacity of each layer is calculated as:

$$WL = \frac{WWM}{\left(\theta_{fc} - \theta_{wp}\right)} \tag{9}$$

Where $WL$ is the soil profile thickness matrix of tension water, $WL = (WUL, WLL, WWL)$, representing the thickness of the upper, lower, and entire soil profile of tension water in mm, respectively, and $WWM = (WUM, WLM, WM)$. Similarly, the thickness of the conceptual soil profile of free water is calculated as:

$$SL = \frac{SM}{\left(\theta_s - \theta_{fc}\right)} \tag{10}$$

Subsequently, linear interpolation is used to adjust the IN-SITU data and CLDAS reanalysis soil moisture data, both at varying depths, to match the thickness of the conceptual soil profile. This step is followed by the calculation of tension and

free water storage, derived from the transformed IN-SITU and CLDAS data. The calculation formula is as follows:

$$\begin{cases} WOB_i = \left(\theta\_WL_i - \theta_{wp}\right) \times WL' \\ \\ WL' = \begin{pmatrix} WUL & 0 & 0 \\ 0 & WLL & 0 \\ 0 & 0 & WWL \end{pmatrix} \end{cases} \tag{11a}$$

$$SOB_i = \left(\theta\_SL_i - \theta_{fc}\right) \times SL \tag{11b}$$

Where, $SOB$ and $WOB = (WUOB, WLOB, WWOB)$ respectively represent the free water storage and tension water storage at various layers, derived from observation data. These are referred to as the observed free water storage and observed tension water storage. $\theta\_WL = (\theta\_WUL, \theta\_WLL, \theta\_WWL)$ and $\theta\_SL$ indicate the soil moisture contents after linear interpolation to the respective conceptual soil profile thicknesses. The subscript $i$ indicates different data sets, namely IN-

SUIT or CLDAS.

Finally, using the WKNN method, soil moisture data from the IN-SUIT dataset is integrated with the CLDAS dataset. The specific implementation steps are as follows:

(1) Normalize the observed free water content and observed tension water content from the dataset using the min-max normalization method. Denote the normalized observation vector as $PSM$:

$$PSM_i = (WOB_i', SOB_i') \tag{12}$$





(2) The Minkowski distance is used to measure the proximity between the IN-SUIT data under evaluation and historical

samples. A smaller distance indicates a closer match between the evaluated soil moisture content and the historical sample.

The distance is calculated as follows:

$$d = \left( \sum_{j=1}^{n} |psm_{\text{IN-SUIT},j}^{RTD} - psm_{\text{IN-SUIT},j}^{HD}|^p \right)^{1/p} \tag{13}$$

Where, $psm_{\text{IN-SUIT},j}$ represents the jth element of the vector $\boldsymbol{PSM}_{\text{IN-SUIT}}$; $n$ is the dimension of $\boldsymbol{PSM}_{\text{IN-SUIT}}$. Superscript

*RTD* stands for the data under evaluation, and *HD* denotes historical data. The distances between the data under evaluation

and each historical sample are ranked in ascending order. The *K* nearest historical samples are then selected as reference

indices based on this principle.

(3) The inverse distance weighting method is used to calculate the final observed free water storage and observed tension

water storage based on the *K* nearest historical samples:

$$\omega_m = \frac{1}{d_m} \bigg/ \sum_{m=1}^{K} \frac{1}{d_m} \tag{14a}$$

$$\boldsymbol{PSM}_{RGC} = \sum_{m=1}^{K} \omega_m \boldsymbol{PSM}_{CLDAS,m} \tag{14b}$$

Where, $\boldsymbol{PSM}_{RGC}$ is the normalized merged observational soil moisture vector; ω represents the inverse distance weights;

$\boldsymbol{PSM}_{CLDAS,m}$ is the normalized CLDAS observation data vector corresponding to the *m-th* sample. The merged observed

tension water storage $\boldsymbol{WOB}_{RGC} = (WUOB_{RGC}, WLOB_{RGC}, WWOB_{RGC})$ and merged observed free water storage $SOB_{RGC}$ are

obtained after denormalization

In this study, the Grid Search (GS) method (Bergstra and Bengio, 2012; Alibrahim and Ludwig,2021) is employed to

optimize the hyper-parameters *K* and *p*, accompanied by a three-fold cross-validation. This approach ensures maximum R-

squared and minimum root mean squared error for the test set, balancing model generalizability with accuracy (Table S2-1).

For the multi-source soil moisture data fusion, 70% of the historical dataset is used as the training set for model training,

while the remaining 30% serves as the test set to verify model generalization.

## 2.5 Evaluation metrics

In this study, we use four metrics to assess the assimilation effectiveness, focusing on both optimal single-value and

ensemble performances, as suggested by McInerney et al. (2020). The optimal single-value performance, indicating the

highest simulation accuracy, is represented by the ensemble mean values of the simulated discharge. The ensemble

performance evaluation, in contrast, examines the simulated discharge ensemble through the lens of ensemble forecasting,

covering both the overall performance of ensemble and its reliability.





For quantitatively assessing the optimal single-value performance, we employ the Normalized Nash-Sutcliffe efficiency
coefficient (NNSE) (Nossent and Bauwens, 2012) and the root mean squared error (RMSE). The Continuous Ranked Probability Score (CRPS), introduced by Hersbach (2000), measures the overall performance of ensemble. The reliability component of CRPS, denoted as RELI, focuses on assessing ensemble reliability. For these metrics, we use the ratios of AEnKF to Open Loop (ensemble run without assimilation), represented as $R_{RMSE}$, $R_{CRPS}$, and $R_{RELI}$. Moreover, the event-averaged values of these ratios are denoted as $MR_{RMSE}$, $MR_{CRPS}$, and $MR_{RELI}$. The mean value of NNSE for multiple flood
events is denoted as MNNSE. In synthetic cases, 'synthetic true values' serve as the benchmark for all evaluation metrics, while observed values are used in real-world cases. Additional information about these metrics can be found in Supplement 3.

## 3 Study areas and data

The Wuqiangxi catchment (Fig. 2), is located in the middle reaches of the Yuan River, the third-largest tributary of the
Yangtze River. It covers an area of approximately 8,033 km², with elevations ranging from 42 to 1,396 meters. The geographical coordinates of the catchment extend from 109°44′ E to 111°01′ E and from 28°01′ N to 29°07′ N. Situated in the mid-subtropical monsoon humid climate zone, Wuqiangxi catchment experiences abundant rainfall and rich water resources. The average annual precipitation is around 1,400 mm, with uneven distribution throughout the year, predominantly during the flood season (March to September). The catchment, located in the subtropical evergreen and
deciduous broadleaf forest belt, features dense vegetation, predominantly forests and grasslands. The soil texture is primarily loamy. For this study, the Wuqiangxi Catchment is divided into 10 sub-basins, each identified by red underlined numbers in Fig. 2b, ensuring at least one rain gauge in each sub-basin. For an overview of the data used in this study, please see Supplement 4.

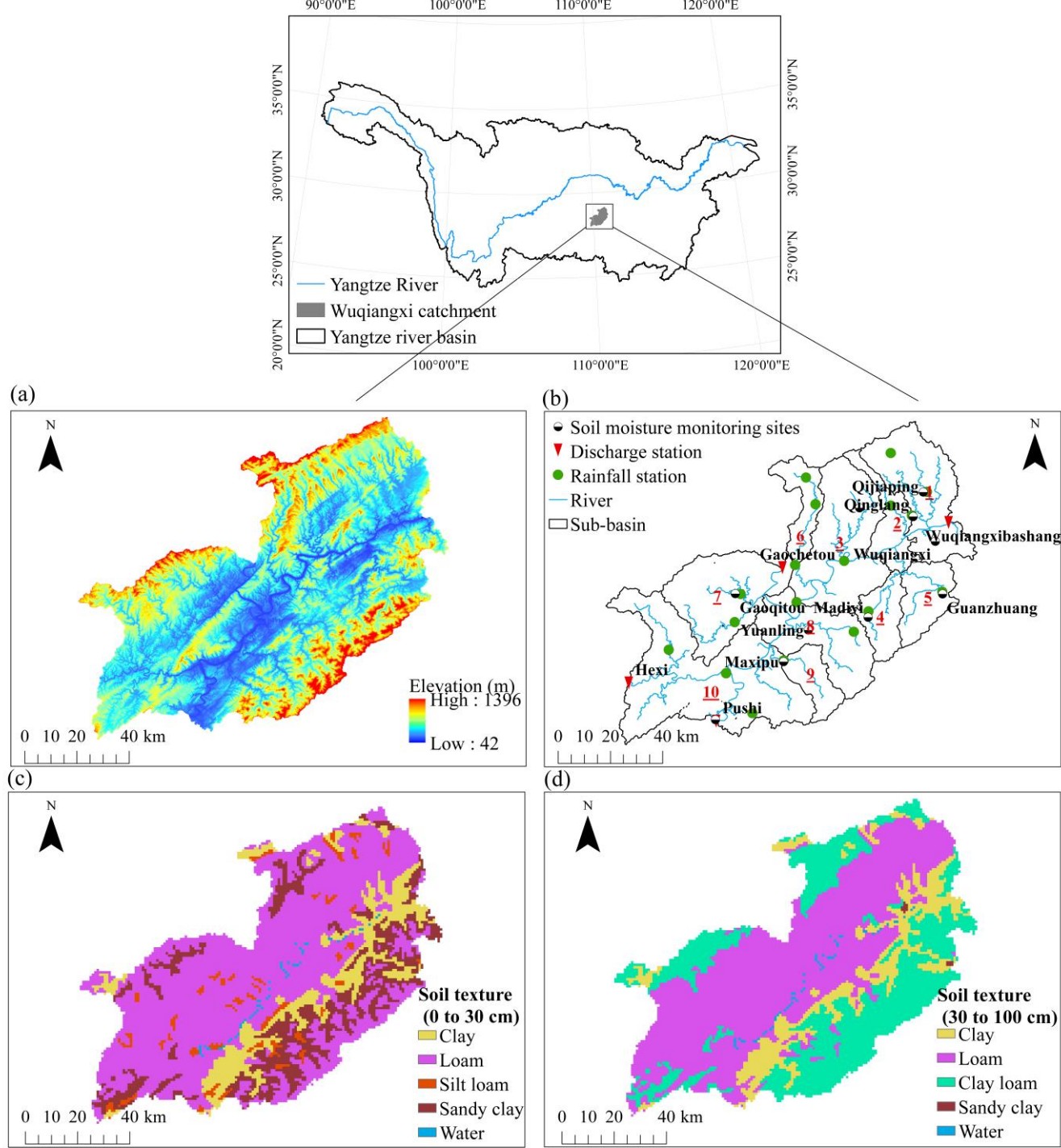

**Fig. 2 Study catchment. (a) Digital Elevation Map (DEM); (b)Sub-basins and observation stations; (c) Soil texture (0 to 30 cm); (d) Soil texture (30 to 100 cm).**





# 4 Experimental setup

## 4.1. Warming-up period

In China, hydrometeorological data are typically reported at sub-daily intervals during flood periods and on a daily basis
otherwise, to support flood forecasting and water resource management. The Xin'anjiang model operates in two modes to
meet these needs: it uses hourly simulations for flood forecasting and daily simulations for managing water resources. The
hourly simulations require initial soil moisture for each sub-basin, which are derived from the daily simulations (Chen et al.,
2023). Consequently, a daily simulation must be performed prior the hourly simulation, starting at least four months earlier
to allow for a sufficient warming-up (spin-up) period. This period enables the soil moisture simulated daily, driven by
observed hydrometeorological data, to gradually approaches actual soil moisture (Kim et al., 2018). As long as the warming-
up period is adequately long, the influence of initial soil moisture on the simulation at the end of warming-up period,
allowing soil moisture for daily simulation to be used as initial conditions for hourly simulation (Yao et al., 2012).

## 4.2. Synthetic cases

In the synthetic cases, the hydrological model operates on an hourly timestep with a maximum lead time of 24 hours, and
ensemble simulations involve 100 members. The initial soil moisture is set to half of the maximum value. To capture peak
flows even at the maximum lead time, the start of each flood event is advanced by 24 hours. However, due to the lack of
hourly observations prior to the actual onset of the flood, data for these initial 24 hours are derived by interpolating from
daily observations. Synthetic data are generated as follows. Firstly, historical flood events are utilized to apply the MAP
method, producing an optimal hyperparameters set $\widehat{\boldsymbol{\psi}}$. Here, $\hat{\sigma}_{lnp}$ and $\hat{\alpha}_{lnp}$ control the error model of forcing data
(Supplement 1.1). This introduces random perturbations into hourly rainfall observations, creating a set of random rainfall
data, referred to as 'synthetic true rainfall'. Similarly, $\hat{\sigma}_{yd}$ and $\hat{\alpha}_{yd}$ manage the observation error model (Supplement 1.2),
perturbing basin inflow to produce a dataset known as 'synthetic true inflow'. Subsequently, the Xin'anjiang model, driven
by the 'synthetic true rainfall' and 'synthetic true inflow', along with the recommended parameters (see Table 1), outputs
state variables (such as tension water storage) and discharge at the catchment outlet for each timestep. These outputs are
designated as the 'synthetic true state variables' and 'synthetic true discharge'. In the final phase, optimal hyperparameter
sets $(\hat{\sigma}_{ys}, \hat{\alpha}_{ys})$ and $(\hat{\sigma}_{yd}, \hat{\alpha}_{yd})$ are applied to the observation error model, respectively. This step introduces random
perturbations into the 'synthetic true state variables' and 'synthetic true discharge', resulting in the creation of 'synthetic
observed state variables' and 'synthetic observed discharge'. Specifically, synthetic observations of tension and free water
storage are employed to update the simulated values in the Xin'anjiang model. On the other hand, synthetic discharge
observation is utilized for updating cumulative channel flow. Both these assimilation processes are conducted at an hourly
interval.





### 4.3. Real-world cases

In the real-world cases, the timestep and number of ensemble members are the same as in the synthetic cases. However, the maximum lead time is set to 8 hours to avoid missing peak flows. The observational tension water storage $\boldsymbol{WOB}_{RGC}$ and free
water storage $SOB_{RGC}$, as introduced in Section 2.4, are used to assimilate the simulated tension and free water storage in the Xin'anjiang model, with an assimilation interval of 8 hours. Additionally, discharge observation is assimilated into the cumulative channel flow with a 1-hour interval. Note that in both the synthetic and real-world cases in this study, we use historical rainfall data as a perfect proxy for rainfall prediction with the aim of assessing temporal persistence of the assimilation effect without introducing uncertainty from numerical weather prediction. Temporal persistence refers to the
duration over which the updating applied to state variables by AEnKF at the start of forecasting continue to hold in the future. By introducing the unbiased perturbations into the model forcing and states, and running the Xin'anjiang model in ensemble mode without assimilation, the operation is referred to as Open Loop (OL). In contrast, an ensemble run integrated with the AEnKF assimilation is referred to as AEnKF. To reduce the effects of random perturbations on outcomes, each flood event in our study is subjected to five repeated ensemble simulations. We then select the simulation corresponding to the median
RMSE in the forecasted discharge as our final outcome.

### 5. Results and discussion

### 5.1. Synthetic Cases

### 5.1.1. Hyperparameter estimation for error models

Most current assimilation methods, while suboptimal for complex hydrological processes, still yield reliable outcomes within
a reasonably characterized uncertainty. Our approach to error characterization, widely adopted in hydrology, involves perturbing model forcing, observations, and states from an assumed distribution. We applied the MAP method for global hyperparameter optimization, with optimal parameters detailed in Table 2. These optimized hyperparameters are used in error models for both synthetic and real-world cases. However, given the limited flood events used for calibration, the hyperparameter optimization, akin to model parameter calibration, might exhibit uncertainty and parameter equifinality,
leading to multiple hyperparameter combinations may produce similar ensemble simulations.

**Table 2. Hyperparameter estimated by MAP method**

| Hyperparameter | Optimal value |
|---|---|
| $\sigma_{ys}$ | 0.108 |
| $\alpha_{ys}$ | 0.340 |
| $\sigma_{yd}$ | 0.106 |
| $\alpha_{yd}$ | 0.312 |
| $\sigma_{lnp}$ | 0.482 |





| | |
|---|---|
| $\alpha_{lnp}$ | 0.456 |
| $\sigma_s$ | 0.058 |
| $\sigma_d$ | 0.220 |

## 5.1.2. The time window of AEnKF

The AEnKF employs observational data from both the current and preceding time periods for assimilation, with the duration of the past interval defined by the time window, $\omega$. Determining the optimal duration for assimilating past observations is critical for the effectiveness of AEnKF. If the time window is set too narrowly, the system might fail to fully capitalize on historical data to enhance assimilation precision. On the other hand, an excessively broad time window could lead the nonlinear system to incorporate irrelevant information from distant past periods, potentially undermining assimilation performance. Therefore, we conducted tests to assess the impact of varying time windows on discharge forecast accuracy. Specifically, for soil moisture observations, we explored three different time windows: $\omega_s = 1$ hour, $\omega_s = 3$ hours, and $\omega_s = 5$ hours. Similarly, for discharge observations, we examined time windows $\omega_d$ of 1 hour, 3 hours, and 5 hours. To facilitate clarity, these assimilation time windows are denoted using dual numerical subscripts. For instance, the AEnKF utilizing $\omega_s = 1$ and $\omega_d = 3$ is designated as AEnKF$_{13}$, and similar nomenclature applies to other configurations.

The disparity in forecast discharge accuracy across different time windows is presented in Fig. 3. It shows the MNNSE and MR$_{RMSE}$ metrics for forecast discharge across lead times of 1 to 24 hours under assorted time window combinations. It's observed that the performance of the AEnKF varies across these time windows. The most effective assimilation across all lead times is achieved with $\omega_s = 3$ and $\omega_d = 3$. It's important to note, however, that even with the least effective time windows ($\omega_s = 1$ and $\omega_d = 5$), performance of AEnKF still surpasses that of the EnKF. In more detail, the time windows for soil moisture and discharge have complex interactions that collectively influence the forecast results for catchment outlet discharge. For soil moisture assimilation, a 3-hour window demonstrates the most significant benefits. The 5-hour window outperforms the 1-hour in most cases, except when $\omega_d = 3$, where the reverse is true. In the assimilation of outlet discharge, the 3-hour window generally proves most effective, but with a larger soil moisture window ($\omega_s = 5$), assimilating discharge data with 1-hour window yields the best results. Almost universally, the 1-hour window performs as well as or surpasses the 5-hour window. This indicates that longer assimilation windows do not necessarily yield better results. This may be due to the nonlinearity of the hydrological model, where overly long windows can result in the system assimilating excessive noise, which negates the benefits derived from incorporating past observations. Therefore, for upcoming studies involving synthetic data cases, the AEnKF will utilize assimilation time windows of $\omega_s = 3$ hours and $\omega_d = 3$ hours.





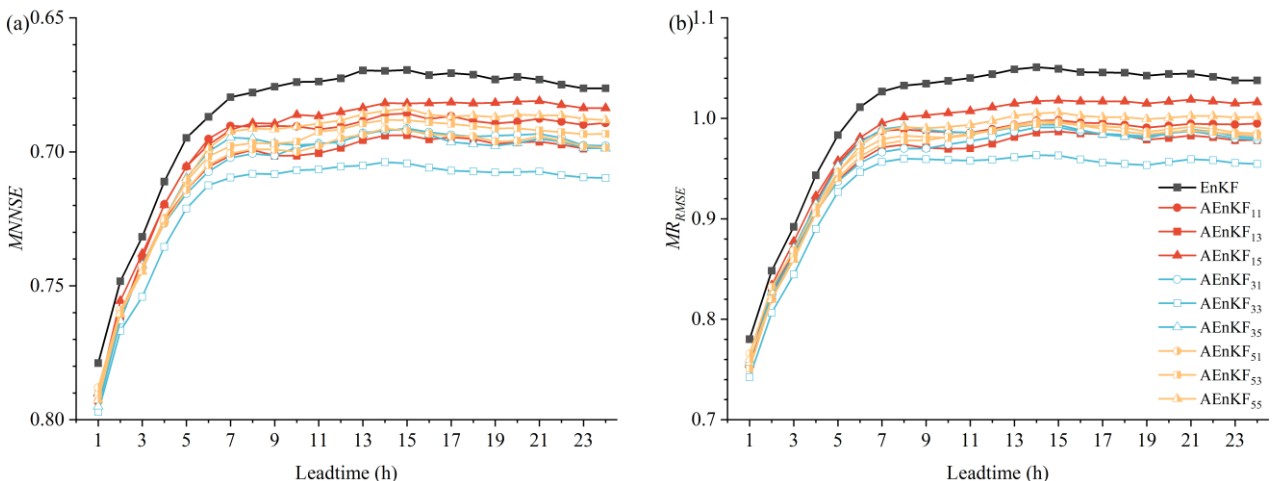

**Fig. 3. The forecasted discharge accuracy under various time windows. (a)MNNSE, (b) MR$_{RMSE}$.**

### 5.1.3. Multivariate observation assimilation scheme

Upon determining the assimilation time window, we meticulously analyzed the variances between three unique AEnKF assimilation strategies. These include the assimilation of solely observed soil moisture (labeled as AEnKF$_S$), the assimilation of solely observed outlet discharge (labeled as AEnKF$_Q$), and a joint assimilation of both two observations types (labeled as AEnKF$_{SQ}$).

**One-step prediction**

In our assessment of one-step prediction of outlet discharge, we examined the optimal single-value performance and ensemble efficacy of the three schemes. The evaluation of the optimal single-value performance was conducted using NNSE and R$_{RMSE}$ as metrics. Fig. 4 (a-f) illustrates the NNSE values during six flood events in 2023 (refer to Table S4-1). Significantly, in events No.2023062100 and No.2023072516, the catchment experienced minimal rainfall (only one hour of rainfall exceeded 3mm), with the flood dynamics largely driven by basin inflows. Consequently, updates to soil moisture

within the catchment had no influence on flood progression, and while assimilating observed discharge data slightly enhanced flood forecasting accuracy, the improvement was minimal and could be considered negligible. Conversely, in the other four events where rainfall predominantly influenced the flood dynamics, all three assimilation schemes outperformed the OL mode in NNSE scores, indicating improvements in one-step prediction accuracy to varying extents. Among these, AEnKF$_{SQ}$, simultaneously assimilating observed soil moisture and discharge data, notably surpassed the other two schemes.

This superiority is further supported by the R$_{RMSE}$ statistics in Fig. 4 (g), where the MR$_{RMSE}$ for AEnKF$_{SQ}$ showed a decrease of 0.12 and 0.11 compared to AEnKF$_S$ and AEnKF$_Q$, respectively. Moreover, soil moisture assimilation and discharge assimilation exhibited comparable performances, with only a marginal reduction of 0.01 in MR$_{RMSE}$.





Fig. 4. The optimal single-value performance of three AEnKF assimilation schemes for synthetic cases. (a-f) NNSE, (g) $R_{RMSE}$.





Subsequently, we conducted an evaluation of the ensemble performance across six flood events, specifically examining the overall ensemble performance as measured by CRPS and the ensemble reliability as indicated by the RELI metric. In Fig. 5 (a), the distribution of $R_{CRPS}$ values are showcased. For $AEnKF_Q$ scheme, $R_{CRPS}$ values fluctuated between 0.8 and 1.01, averaging at 0.89; for $AEnKF_S$, the range is 0.72 to 1.03 with an average of 0.91; for $AEnKF_{SQ}$, it varied from 0.68 to 1.01,

averaging 0.81. This demonstrates an enhancement in overall ensemble performance for all schemes over the OL model, particularly for $AEnKF_{SQ}$, which significantly outshone $AEnKF_S$ and $AEnKF_Q$. However, the difference in $MR_{CRPS}$ between $AEnKF_S$ and $AEnKF_Q$ is minimal. The Fig. 5 (b) illustrates the $R_{RELI}$ scores, showing a similar trend of improved ensemble reliability for all three schemes over OL. Here, the reliability of $AEnKF_{SQ}$ is notably higher than that of both $AEnKF_S$ and $AEnKF_Q$. On the other hand, $AEnKF_Q$ is more reliable compared to $AEnKF_S$.


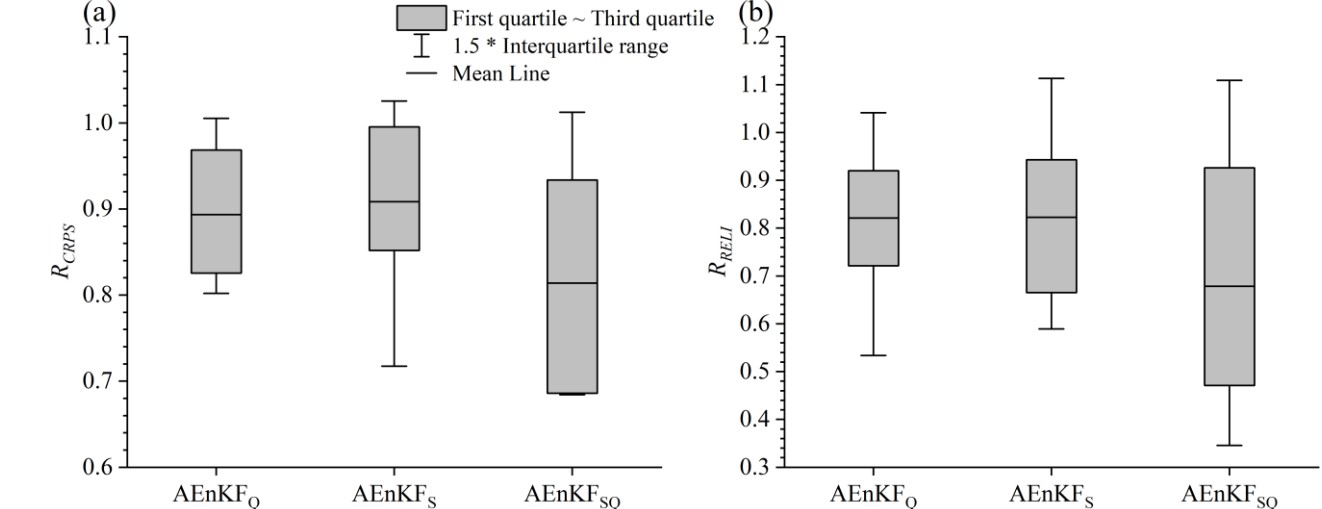

**Fig. 5. The ensemble performance of three AEnKF assimilation schemes for synthetic cases. (a) $R_{CRPS}$, (b) $R_{RELI}$.**

### Impact on state variables

Within the context of one-step prediction, Fig. 6 presents the $R_{RMSE}$ for updated state variables of the Xin'anjiang model under three distinct assimilation schemes, involving free water storage, tension water storage across upper, lower, and total

layers, and cumulative channel flow across all sub-basins. As anticipated, in the $AEnKF_Q$ scheme, which solely updates the cumulative channel flow without involving the runoff generation process, the state variables indicative of soil moisture ($S$, $W$, $WU$, $WL$) remain unaffected. This is reflected in Fig. 6 (a-d), where the mean $R_{RMSE}$ values associated with the grey boxes hover around 1.0. In the context of cumulative channel flow, $AEnKF_Q$ generally achieves a reduction in RMSE relative to the OL.

In the case of $AEnKF_S$, updating the soil moisture impacts both the runoff generation and the subsequent flow routing processes, leading to a response in all state variables. Significantly, $AEnKF_S$ scheme demonstrates the most substantial corrections in free water storage ($S$), consistently yielding lower RMSE values than the OL scheme in all instances. When updating the total tension water storage ($W$), $AEnKF_S$ usually attains lower RMSE values compared to OL. However, this





effect is less marked than that for free water storage. This is illustrated in Fig. 6 (b) where, except for Event 6, the average

values associated with the red boxes exceed those in Fig. 6 (a). Contrastingly, updates to the upper and lower layers of tension water storage in the AEnKF$_S$ scheme produced opposite outcomes, with the RMSE values for these post-updated state variables exceeding those of the OL. This phenomenon can be attributed to the following: Firstly, the initial soil moisture values in this study were set at half of their maximum, and during flood periods, saturation-excess runoff generation mechanism ensures rapid saturation of both the upper and lower tension water, reaching the maximum limits

(*WUM* and *WLM*). Thereafter, due to the physical upper bounds of these variables, the assimilation process is hindered in effectively updating *WU* and *WL* values. Consequently, this may lead to a systematic underestimation of these values compared to actual measurements, and consequently higher RMSE values than OL. In contrast, free water storage, even during flood periods, may not persistently reach its maximum (*SM*), resulting in the most advantageous update effect for it. These results highlight the criticality of choosing appropriate state variables for updates in hydrological model state updating,

particularly when utilizing methods such as AEnKF.

In the case of AEnKF$_{SQ}$, the updates to soil moisture state variables show similarities to those in AEnKF$_S$. However, when it comes to the updates of cumulative channel flow, AEnKF$_{SQ}$ effectively integrates the strengths of both AENKF$_Q$ and AENKF$_S$, resulting in a more outstanding performance. This outcome suggests that the concurrent assimilation of both soil moisture and discharge observations can efficiently utilize the advantages of each, leading to a greater assimilation accuracy

than the assimilation of a single observation source.





**Fig. 6. Effects of three assimilation schemes on state variables. (a) Free water storage (*S*), (b) Tension water storage (*W*), (c) Upper tension water storage (*WU*), (d) Lower tension water storage (*WL*), (e) Cumulative channel flow (*QC*).**



**Discharge Forecasting**

In our assessment, we analyzed the discharge simulation precision of three assimilation schemes over lead times ranging from 1 to 24 hours, aiming to gauge the temporal persistence of the assimilation effect. Fig. 7 (a-f) presents the NNSE for six flood events. The events identified as 2023062100 (Fig. 7 d) and 2023072516 (Fig. 7 f) were mainly driven by inflows, exhibiting only slight improvements in state updates, and therefore are not included in further discussions. For the events identified as No.2023050416 and No.2023052008, the NNSE of each assimilation scheme exceeded that of OL across all

lead times, indicating a consistent assimilation impact lasting up to 24 hours. For the event labeled No.2023040308, the temporal persistence for the three schemes is noted as 8, 8, and 2 hours, respectively; in the case of the event marked as No.2023063000, these durations are 5, 4, and 1 hour. Importantly It is noteworthy that even in No.2023063000, despite being the least effective, the NNSE discrepancy between $AEnKF_{SQ}$ and OL for lead times exceeding 8 hours remains below 0.02. Furthermore, $AEnKF_{SQ}$ demonstrated superior performance in most flood events across all lead times, compared to

both $AEnKF_{S}$ and $AEnKF_{Q}$. The notable exception is event No.2023052008, where $AEnKF_{SQ}$ excelles within a 4-hour lead time but slightly lagged behind the other two schemes beyond this duration. Nevertheless, the variance in NNSE for $AEnKF_{SQ}$ during this event stayed below 0.02. Fig. 7 (g) statistically illustrates the $R_{RMSE}$ values. Notably, the $MR_{RMSE}$ for each of the three assimilation schemes remains below 1.0 for all lead times, signifying that in terms of event averages, each scheme achieves a temporal persistence of up to 24 hours. Additionally, the discharge forecast accuracy across nearly all

lead times is ranked with $AEnKF_{SQ}$ surpassing $AEnKF_{S}$, which itself exceeds $AEnKF_{Q}$.



**Fig. 7. Assessment of forecast discharge accuracy across three assimilation schemes during 1 to 24-hour lead times. (a-f) NNSE, (g) $R_{RMSE}$.**





## 5.2. Real-world Cases

In real-world cases, the sensitive parameters of Xin'anjiang model are calibrated based on historical flood events. After global optimization using the SCE-UA, the Xin'anjiang model, equipped with optimally sensitive parameters, exhibited an average NSE of 0.88 for the calibration events and 0.78 for the validation events, demonstrating reliable and credible flood simulation and forecasting capabilities.

Considering that soil moisture observations are obtained every 8 hours in real-world cases, as opposed to hourly in synthetic cases, we have drawn on synthetic case results to establish an assimilation time window for soil moisture as close as possible to 3 hours, set at 8 hours. Therefore, $\omega_s$ is designated as 8 hours, utilizing only the observations from the current time and those from 8 hours earlier for assimilation. For the discharge assimilation, we set the time window to be consistent with the synthetic cases, i.e., $\omega_d = 3$ hours. Additionally, guided by the insights from synthetic cases, in real-world cases, we incorporate all available soil moisture observations but limit updates to the free water storage component of the Xin'anjiang model.

### 5.2.1. Fusion of in situ data with CLDAS soil moisture data

The soil moisture data fused using the WKNN model exhibits enhanced timeliness, with the soil moisture in each conceptual soil profile aligning closely with that of the CLDAS data (Table 3). Specifically, during the calibration set, the correlation coefficients with CLDAS data consistently exceed 0.9. Additionally, the correlation coefficients in the validation set are 0.85, 0.80, 0.84, and 0.74, respectively, indicating that the WKNN model possesses robustness and generalizability. It effectively captures the statistical relationship between point-scale and areal-scale soil moisture datasets.

**Table 3. The correlation coefficient between the fused soil moisture data and CLDAS soil moisture data.**

| Sub-basin | $WWOB_{RGC}$ | | $WUOB_{RGC}$ | | $WLOB_{RGC}$ | | $SOB_{RGC}$ | |
|---|---|---|---|---|---|---|---|---|
| | Calibration | Verification | Calibration | Verification | Calibration | Verification | Calibration | Verification |
| 1 | 0.99 | 0.95 | 0.98 | 0.83 | 0.94 | 0.94 | 0.98 | 0.80 |
| 2 | 0.95 | 0.94 | 0.98 | 0.76 | 0.95 | 0.94 | 0.98 | 0.68 |
| 3 | 0.97 | 0.69 | 0.87 | 0.74 | 0.97 | 0.71 | 0.93 | 0.74 |
| 4 | 0.98 | 0.85 | 0.86 | 0.75 | 0.93 | 0.72 | 0.98 | 0.63 |
| 5 | 0.98 | 0.75 | 0.98 | 0.88 | 0.98 | 0.86 | 0.90 | 0.64 |
| 6 | 0.97 | 0.82 | 0.88 | 0.76 | 0.83 | 0.80 | 0.92 | 0.77 |
| 7 | 0.98 | 0.87 | 0.97 | 0.89 | 0.98 | 0.89 | 0.90 | 0.71 |
| 8 | 0.99 | 0.98 | 0.95 | 0.95 | 0.98 | 0.97 | 0.98 | 0.91 |
| 9 | 0.98 | 0.89 | 0.76 | 0.74 | 0.84 | 0.81 | 0.80 | 0.79 |
| 10 | 0.88 | 0.72 | 0.78 | 0.66 | 0.86 | 0.72 | 0.89 | 0.77 |
| Average | 0.97 | 0.85 | 0.90 | 0.80 | 0.93 | 0.84 | 0.93 | 0.74 |





### 5.2.2. Multivariate observation assimilation scheme

**Free water storage update**

545 Within the context of one-step prediction, Fig. 8 displays the impact of updating free water storage in three different assimilation schemes, quantified by $R_{RMSE}$. For the AEnKF$_Q$ scheme, there is no update to free water storage, leading to expected $R_{RMSE}$ values oscillating near 1.0. Conversely, both AEnKF$_S$ and AEnKF$_{SQ}$ successfully updated free water states, with mean values of $R_{RMSE}$ for free water storage in flood event simulations lying between 0.48 and 0.74. This demonstrates

550 a 26% to 52% average reduction in RMSE for free water storage across various flood events, in comparison to the OL mode. Moreover, Fig. 8 reveals that, in the vast majority of cases, the whiskers (representing 1.5 times interquartile range) of the red and blue boxes remain below 1.0. This indicates that both AEnKF$_S$ and AEnKF$_{SQ}$ successfully updated the free water storage in most sub-basins for most flood events. The effective updates to free water storage will further impact the discharge process at the catchment outlet, which will be discussed in detail in the subsequent sections.

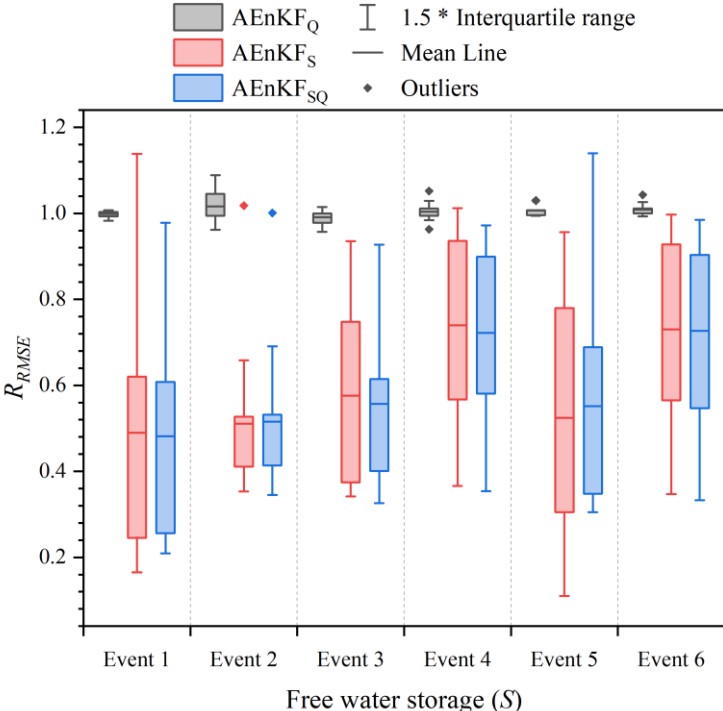

**Fig. 8. Effects of three assimilation schemes on free water storage ($S$).**

**One-step prediction**

We evaluate the optimal single-value performance of AEnKF in one-step (one hour) prediction. Fig. 9 illustrates the NNSE and $R_{RMSE}$ values achieved through three AEnKF assimilation schemes. In the OL, the mean NNSE stands at 0.76.

560 Following assimilation with three schemes, the mean values of NNSE improve to 0.81, 0.80, and 0.83, respectively. The $R_{RMSE}$ of AEnKF$_Q$ fluctuates between 0.78 and 1.0, with an average of 0.88; for AEnKF$_S$, it ranges from 0.71 to 1.01,





averaging 0.91; and for AEnKF$_{SQ}$, it varies from 0.64 to 0.99, with an average of 0.84. These results show that all three AEnKF assimilation schemes enhance the optimal single-value performance, with AEnKF$_{SQ}$ outperforming AEnKF$_Q$, which in turn exceeds AEnKF$_S$. Moreover, AEnKF$_{SQ}$ achieves a higher improvement ceiling in certain flood events. For instance, the maximum reduction in RMSE reaches 22% for AEnKF$_Q$, 29% for AEnKF$_S$, and up to 36% for AEnKF$_{SQ}$.

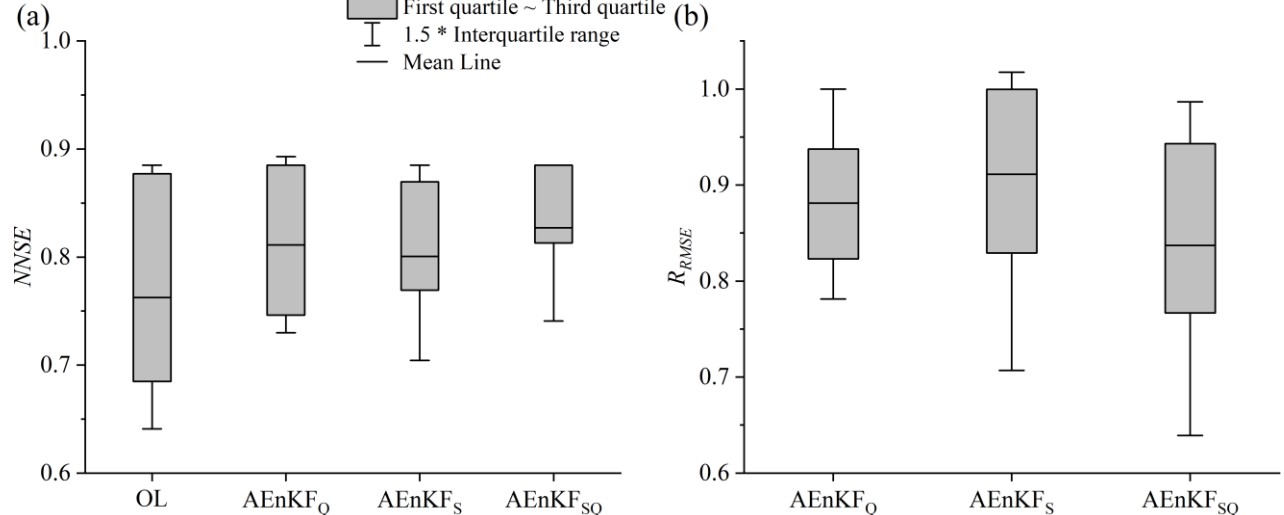

**Fig. 9. The optimal single-value performance of three AEnKF assimilation schemes for real-world cases. (a) NNSE, (b) R$_{RMSE}$.**

Fig. 10 utilizes R$_{CRPS}$ and R$_{RELI}$ metrics to evaluate overall ensemble performance and reliability. The R$_{CRPS}$ values for AEnKF$_Q$ are in the range of 0.81 to 1.0, averaging at 0.89; for AEnKF$_S$, they span from 0.71 to 1.02, averaging 0.91; and for AEnKF$_{SQ}$, they vary from 0.66 to 0.98, averaging 0.85. Notably, AEnKF$_Q$ exhibits the narrowest boxplot, indicating a more focused distribution of R$_{CRPS}$ for this scheme. The average R$_{CRPS}$ for AEnKF$_S$ closely aligns with that of AEnKF$_Q$, yet its boxplot shows greater breadth at both the top and bottom, suggesting a higher potential for improvement in overall ensemble performance but with increased instability. In contrast, the average R$_{CRPS}$ for AEnKF$_{SQ}$ is lower than those of the first two. While the boxplot width for AEnKF$_{SQ}$ is similar to that of AEnKF$_S$, the upper boundary of the boxplot aligns more closely with AEnKF$_Q$, and the upper whisker is shorter than that of AEnKF$_Q$, indicating a comprehensive superiority of AEnKF$_{SQ}$ in overall ensemble performance compared to both AEnKF$_S$ and AEnKF$_Q$. Similar findings also emerge in the assessment of ensemble reliability. AEnKF$_S$ and AEnKF$_Q$ exhibit similar mean R$_{RELI}$ values, but the boxplot for AEnKF$_Q$ is more constricted. In contrast, AEnKF$_{SQ}$ shows a thorough superiority in ensemble reliability compared to both AEnKF$_S$ and AEnKF$_Q$.





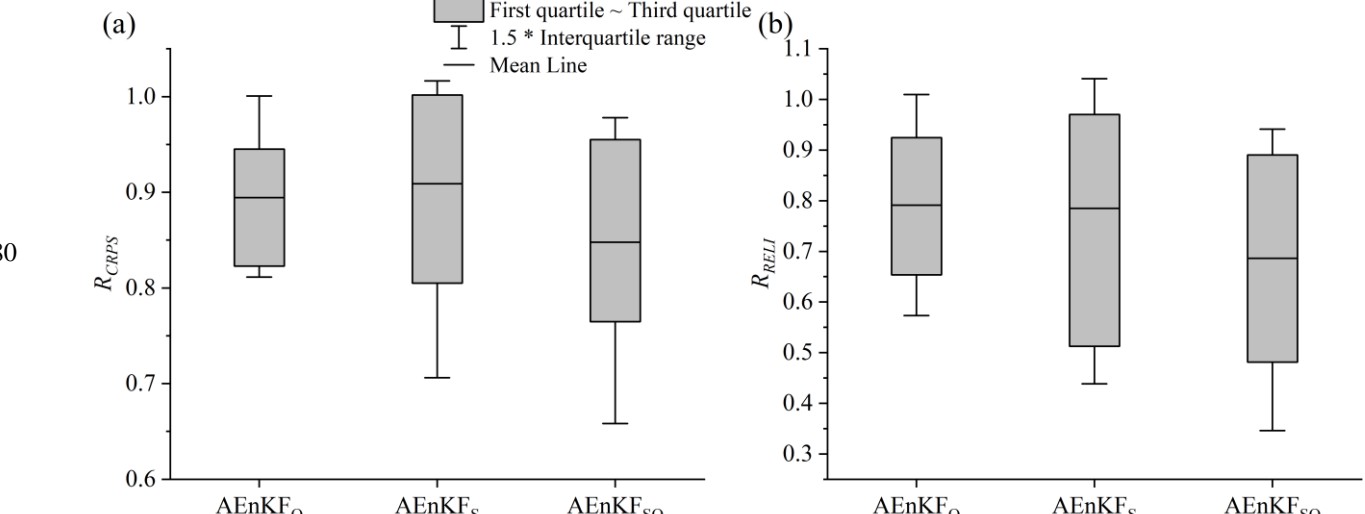

**Fig. 10. The ensemble performance of three AEnKF assimilation schemes for real-world cases. (a) R_CRPS, (b) R_RELI.**

### Discussion of two flood events

In flood simulation and forecasting, peak flow rates are a primary focus for researchers. Using the two flood events with the most significant peak flow errors in the OL mode (No.2023040308 and No.2023052008) as case studies, we examined the variations in free water storage and discharge at the catchment outlet.

Fig. 11 display the hydrographs simulated for No.2023040308. Black lines (dots) signify observed values. Grey lines and bands represent the ensemble mean and range of the OL, respectively. Similarly, green lines and bands illustrate the ensemble mean and range for the AEnKF. In examining the time series of free water storage, it is evident that observational data points almost never fall within the grey bands of the OL scheme. This indicates a notable difference between the soil moisture levels simulated by the Xin'anjiang model and those derived from observational data. Both $AEnKF_S$ and $AEnKFS_Q$ exhibit similar update patterns, where the post-update ensemble mean values significantly shift towards observational data. Concurrently, this adjustment expands the ensemble bands, indicating an increase in ensemble simulation accuracy for $AEnKF_S$ and $AEnKFS_Q$, along with an increased ensemble spread. In the analysis of the discharge time series, it becomes evident that the ensemble distribution from the AEnKF aligns more closely with observational data and presents a narrower bandwidth than that of the OL. This trend suggests that the ensemble accuracy with AEnKF exceeds that of the OL scheme, and also demonstrates a reduced ensemble spread. Furthermore, the ensemble distribution observed during peak periods is more expansive than during the onset and recession periods of flood. This is attributed to the error models applied. These models introduce larger perturbations in the assimilation system during peak periods, leading to a broader ensemble distribution, which, in turn, ensures a more effective assimilation during these critical periods. In examining the time series of discharge, it is noted that both $AEnKF_Q$ and $AEnKF_S$ significantly reduced the height of the simulated flood peak. The $AEnKF_Q$ scheme shows effectiveness around the 20th hour, following the assimilation of approximately 20 discharge observations, achieving a relative error of 17% in the simulated flood peak (maximum instantaneous flow) compared to the



observed peak. AEnKF$_S$ started effectively updating the discharge following the assimilation of the third group of soil moisture observations at the 17th hour, which led to a flood peak relative error of 13%. The AEnKF$_{SQ}$ scheme successfully amalgamates the strengths of both, culminating in a reduced flood peak relative error of merely 8%.

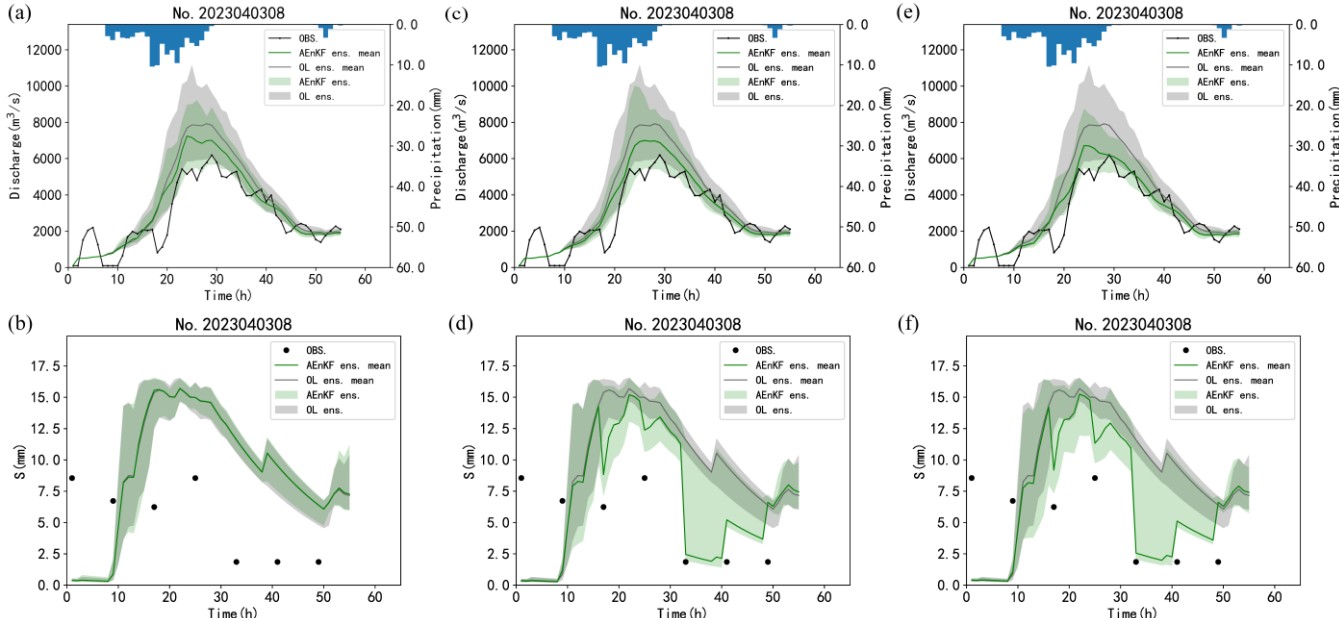

**Fig. 11. Hydrograph during flood event labeled No.2023040308. (a-b) AEnKF$_Q$ Scheme, (c-d) AEnKF$_S$ Scheme, (e-f) AEnKF$_{SQ}$ Scheme. The upper panel shows the discharge at the catchment outlet, and the lower panel displays the free water storage in sub-basin 1.**

In the case of flood event labeled No.2023052008, as illustrated in Fig. 12, the time series exhibits a similar pattern to No. 2023040308. The peak flooding occurred between the 25th and 33rd hours, which corresponds to the period between the fourth and fifth sets of soil moisture observations. During this interval, there is a notable and rapid increase in free water storage. Fig. 12 (c) indicates that the AEnKF$_S$ fails to effectively adjust the discharge volumes around the peak period. Conversely, the AEnKF$_Q$ scheme, which focused on updating cumulative channel flow, successfully rectified the peak flooding. Owing to the ineffectiveness of free water content updates in discharge correction, the assimilation impact of AEnKF$_{SQ}$ closely matched that of AEnKF$_Q$. In summary, it is apparent that AEnKF$_{SQ}$ effectively integrates the strengths of both the AEnKF$_S$ and AEnKF$_Q$ schemes. Even when one of these strategies fails to update effectively, AEnKF$_{SQ}$ still manages to enhance the precision of discharge predictions.





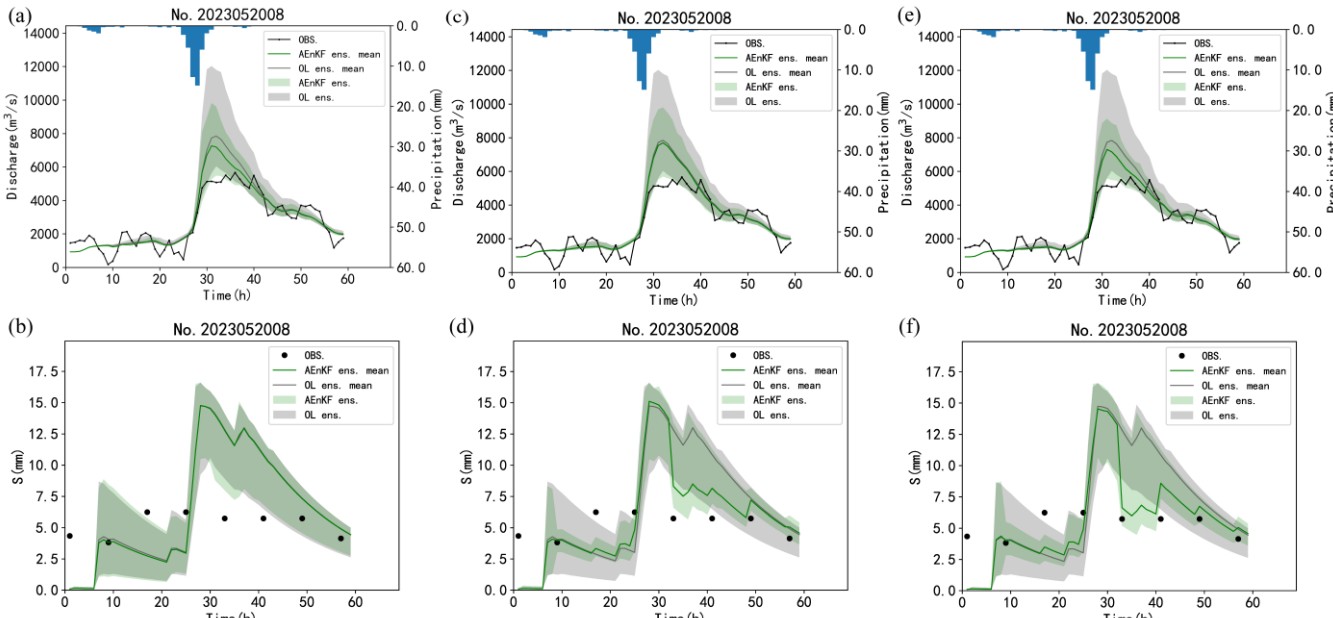

**Fig. 12. Hydrograph during flood event labeled No.2023052008. (a-b) AEnKF$_Q$ Scheme, (c-d) AEnKF$_S$ Scheme, (e-f) AEnKF$_{SQ}$ Scheme. The upper panel shows the discharge at the catchment outlet, and the lower panel displays the free water storage in sub-basin 1.**

**Temporal persistence of the assimilation effect**

In previous analyses, the performance of three assimilation schemes in one-step prediction received attention. This section extends the examination to the temporal persistence of assimilation effects for these schemes. Fig. 13 displays discharge forecasting accuracy across various lead times, as measured by NNSE and $R_{RMSE}$. Within a lead time range of 1 to 8 hours, both AEnKF$_S$ and AEnKF$_{SQ}$ demonstrate improvements in forecasting performance: AEnKF$_{SQ}$ exceeds AEnKF$_S$ within a 5-hour lead time; beyond 6 hours, the accuracy of both becomes similar. AEnKF$_Q$ shows significantly shorter temporal persistence than the other two, slightly outperforming AEnKF$_S$ in one-step prediction but with a rapid decline in accuracy as lead time increases. Past a 5-hour lead time, the assimilation effect of AEnKF$_Q$ vanishes, leading to accuracy slightly below OL. And at different lead times, AEnKF$_{SQ}$ consistently outperforms AEnKF$_Q$. This reveals that employing AEnKF for updating cumulative channel flow may notably enhance discharge forecasting accuracy in shorter lead times. While updating free water storage may not be as effective as AEnKF$_Q$ initially, it ensures a longer-lasting assimilation impact. The scheme of AEnKF$_{SQ}$ merges these strengths, offering robust discharge corrections and an extended temporal persistence of assimilation effects.





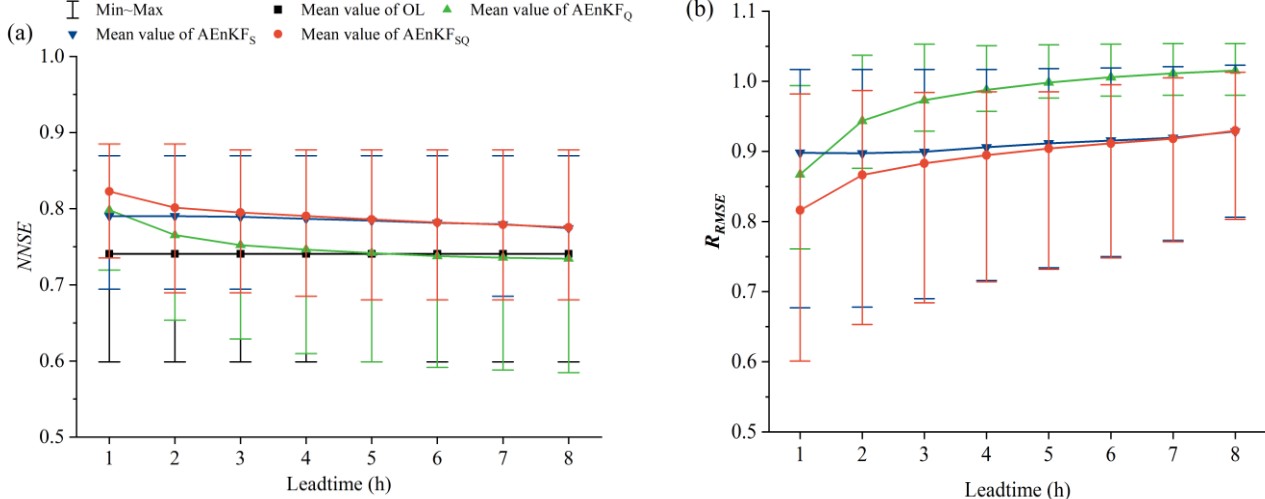

**Fig. 13. The accuracy of forecasted discharge under different lead time. (a) NNSE, (b) $R_{RMSE}$.**

## 6. Conclusions

This study uses the Asynchronous Ensemble Kalman Filter (AEnKF) with enhanced error models for assimilating two types

of observational data into the Xin'anjiang model. The data include observed discharge at catchment outlet and soil moisture gathered from multiple sources. The objective is to diminish error accumulation in the initial conditions of the Xin'anjiang model at the start of flood forecasting, thereby enhancing initial conditions. The assimilation framework includes advanced error models, such as the BGEM model to reduce systematic biases from perturbed soil moisture and the MAP method for the objective estimation of hyperparameters in the error model. The study specifically contrasts three AEnKF assimilation

strategies: (1) The AEnKF$_Q$ scheme updates cumulative channel flow in the Xin'anjiang model by assimilating observed outlet discharge; (2) The AEnKF$_S$ scheme focuses on updating soil moisture variables in the model by assimilating fused soil moisture observations; (3) The AEnKF$_{SQ}$, a joint assimilation scheme, combines both discharge and soil moisture assimilation processes.

Generally, the AEnKF is considered an effective approach for updating hydrological model states. It integrates a greater

amount of observational data while barely increasing the computational burden, making it highly suitable for flood forecasting. The effectiveness of assimilation with the AEnKF relates to the assimilation time window. Results of synthetic data cases indicate that an appropriate setting involves a 3-hour time window for assimilating observed soil moisture and outlet discharge. Moreover, in lead times ranging from 1 to 24 hours, this method consistently outperforms the EnKF approach.

In synthetic case studies, while updating soil moisture state variables of the Xin'anjiang model, it is observed that effective updates are limited to free water storage and total tension water storage. This underscores the significance of choosing appropriate state variables for updates in the application of the AEnKF method. Further analysis revealed that with high-

quality, hourly available observational data, all three assimilation schemes maintained their effectiveness for up to 24-hour lead time. Notably, $AEnKF_{SQ}$ demonstrated enhanced optimal single-value performance, overall ensemble performance, and
ensemble reliability, surpassing both $AEnKF_S$ and $AEnKF_Q$.

In the real-world case studies, we merged soil moisture data from in-suit monitoring sites with the near-real-time CLDAS soil moisture data. This fusion produces spatially distributed data characterized by high temporal immediacy while addressing the limitation of point-scale in in-suit soil data. Contrasting with experiments using synthetic data, extending soil moisture observation intervals to 8 hours impacts the performance of the $AEnKF_S$ scheme. In one-step prediction, the
$AEnKF_{SQ}$ scheme exhibits the highest level of accuracy. Concurrently, the simulation precision of the $AEnKF_Q$ scheme exceeds that observed in $AEnKF_S$. Variations in results are observed under different lead times. $AEnKF_{SQ}$ and $AENKF_S$ consistently demonstrate an assimilation effect duration of 8 hours, in contrast to the 5-hour temporal persistence of assimilation effect of $AENKF_Q$. The use of AEnKF for updating cumulative channel flow markedly enhances the accuracy of discharge forecasting in a brief lead time. In contrast, the adjustment extent of discharge by updating free water storage in a
single-step forecast might be less than that achieved with $AEnKF_Q$. Nevertheless, it guarantees a more sustained assimilation effect. The $AEnKF_{SQ}$ integrates the strengths of the previous two strategies, thereby improving discharge forecasting accuracy even when a particular strategy does not update effectively and prolonging the temporal persistence of the assimilation effect.

**Code/Data availability**

Data will be made available on request.

**Author contribution**

**Junfu Gong**: Conceptualization, Methodology, Software, Visualization, Writing - original draft, Funding acquisition. **Xingwen Liu**: Methodology, Software. **Cheng Yao**: Data curation, Software, Funding acquisition, Writing - review & editing. **Zhijia Li**: Project administration, Funding acquisition, Writing - review & editing. **Albrecht Weerts**:
Conceptualization, Writing - review & editing. **Qiaoling Li**: Supervision, Writing - review & editing. **Satish Bastola**: Writing -review & editing. **Yingchun Huang**: Validation, Funding acquisition. **Junzeng Xu**: Supervision, Resources.

**Competing interests**

At least one of the (co-)authors is a member of the editorial board of Hydrology and Earth System Sciences.





**Acknowledgements**

This work was supported by Xinjiang Uygur Autonomous Region Key Research and Development Project (Grant No. 2023B02044-2); the Postdoctoral Fellowship Program of CPSF (Grant No. GZC20240377); the Fundamental Research Funds for the Central Universities (Grant No. B240201181 and B240203007); the National Natural Science Foundation of China [Grant numbers 52079035 and 51909059]; Anhui Provincial Natural Science Foundation (Grant No. 2208085US06). We would like to express our heartfelt gratitude to Liaofan Lin for his constructive comments and invaluable assistance in
enhancing the quality of this research. The hydrometeorological data were provided by the Hydrology Bureau of Hunan Province, including evaporation, precipitation and discharge. The code of AEnKF used in this study was developed from the Parallel Data Assimilation Framework (PDFA) (http://pdaf.awi.de/trac/wiki) and OPENDA (www.openda.org). The SCE-UA algorithm implemented via Uncertainty Quantifcation Python Laboratory (UQ-PYL) (http://www.uq-pyl.com). We thank these organizations for granting permission to use their data and software.

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
