# Peer review of "State updating in the Xin'anjiang Model: Joint assimilating streamflow and multi-source soil moisture data via Asynchronous Ensemble Kalman Filter with enhanced Error Models"

_Hydrology and Earth System Sciences, 2024_

## Author Comment (AC4)

**Point-by-point response to Anonymous Reviewer #1**

We would like to sincerely appreciate you for the review of our manuscript "State updating in the Xin'anjiang Model: Joint assimilating streamflow and multi-source soil moisture data via Asynchronous Ensemble Kalman Filter with enhanced Error Models" and the constructive suggestions. We sincerely believe these comments facilitate the quality improvement of this manuscript. All the comments have been considered and a point-by-point response has been provided below.

The point-by-point response is formatted as follows:

- Reviewer's comments are shown in blue
- Authors' response are shown in black
- Authors' changes in the manuscript are shown in red. The line numbers indicated in this response are those in the "Revised Manuscript with no changes marked" document
- The unchanged parts of the manuscript are shown in black
* * *
This study provides a comprehensive review of hydrological data assimilation for flood simulation (forecasting). It attempts to integrate soil moisture data from various sources and jointly assimilate them with runoff observations into a hydrological model. The uniqueness of this paper lies in its first-time application of the Asynchronous Ensemble Kalman Filter (AEnKF) for such joint assimilation, with a consideration of the temporal correlation of observation errors. The paper is well-structured, rich in content, and the results are presented clearly, which made it an engaging read for me. Overall, it is a well-conducted study. However, there are some areas that could be further improved, such as the insufficient discussion of the AEnKF method in the introduction. Below are some of my comments and suggestions:

**Response**: Thank you very much for your concise paper summary and positive feedback on our research. We are honored that our paper has captured your interest. We have carefully considered all of your comments and responded them in the subsequent specific comments section.

**Specific Comments:**

============

1. The Asynchronous Ensemble Kalman Filter (AEnKF) is a simple yet effective data assimilation method, well-suited for state updating in hydrological models. However, the authors have not sufficiently discussed the existing research and applications of the AEnKF method. I recommend that the authors emphasize this discussion more prominently in the Introduction (page 3).

**Response**: Thanks for the helpful suggestion. We have recognized this issue and included a discussion on existing research and applications of the AEnKF method in the Introduction section of the revised manuscript (LINES 77-90).

**Revised Manuscript LINES 77-90:**
The AEnKF technique was first applied by Krymskaya (2013) to the problem of history matching in reservoir engineering. The study revealed that AEnKF outperforms EnKF in parameter estimation and utilizes the data with similar efficiency. The AEnKF is recognized for its simplicity and high computational efficiency, offering significant potential in short-term flood forecasting applications. Despite its promise, the scope of research in this area is relatively limited. Among the few studies conducted, Mazzoleni et al. (2018) evaluated AEnKF assimilation in simplified flow routing models, highlighting its exceptional performance in both lumped and distributed flow routing. Tao et al. (2016) summarized the hydrological forecasting test conducted during the 2014 IPHEx-IOP campaign, proposing a framework for improving flood prediction in mountainous regions through the assimilation of discharge data using the AEnKF method, with a focus on enhancing forecast accuracy and reducing uncertainty. In addition, Rakovec et al. (2015) and our earlier study (Gong et al., 2024) applied the AEnKF to the distributed HBV-96 model and the Xin'anjiang model, respectively. These studies examined effectiveness of AEnKF in real-time correction through the assimilation of observed discharge in distributed and semi-distributed hydrological models, revealing that AEnKF outperforms the standard EnKF. However, these studies assimilate only a single type of observational data (e.g., observed discharge) using the AEnKF method, which does not take full advantage of the AEnKF.

2. The discussion of the advantages of AEnKF should be included in the introduction rather than in the methodology section (page 5, Lines 160-163).

**Response**: Thanks for the helpful suggestion. We have deleted this section in the revised manuscript (LINES 191-192), as the advantages of AEnKF have already been discussed in the Introduction.

**Revised Manuscript LINES 191-192:**

The Asynchronous Ensemble Kalman Filter (AEnKF) represents a straightforward enhancement of the Ensemble Kalman Filter (EnKF), utilizing the same assimilation framework as EnKF.

3. It is very interesting that the study considers the temporal correlation of observation errors and rainfall errors in data assimilation, as most studies assume these errors are independent. Could the authors provide more details on how this was specifically implemented? (page 8, Lines 220-222)

**Response**: Thank you very much for your comment. We addressed the temporal correlation of rainfall and runoff observation errors using a simple first-order autoregressive model. By designing an appropriate first-order autoregressive function, we ensured that the error model, which accounts for temporal correlation, maintains the same mean and standard deviation as the original error model after transformation. Please see S1.1 and S1.2 in the Supplement documentation for the details.

**Revised Supplement Part S1.1:**

**S1.1. Uncertainty in model forcing**

In flood forecasting, the most critical model driving data is rainfall. We used log-normal multiplicative perturbation to characterize rainfall errors (McMillan et al., 2011; DeChant and Moradkhani, 2012; Gong et al., 2023):

$$\boldsymbol{P}_j^o(t_i) = \boldsymbol{\delta}^{\boldsymbol{P}}(t_i) \cdot \boldsymbol{P}(t_i) \tag{S1-1}$$

Where $\boldsymbol{P}(t_i) = [P_1(t_i), \dots, P_{N_p}(t_i)]^T \in \mathcal{R}^{N_p}$ is the rainfall observation vector; $N_p$ is the dimensionality of the rainfall observations; $\boldsymbol{\delta}^{\boldsymbol{P}}(t_i)$ is lognormal perturbation matrix. The errors in the precipitation measurement are assumed to be spatially independent, so that, $\boldsymbol{\delta}^{\boldsymbol{P}}(t_i)$ is also a diagonal matrix. The diagonal element is $\delta_n^P(t_i), (n = 1, \dots, N_p)$, and $ln\,\delta_n^P(t_i) \sim N(\mu_{lnp}, \sigma_{lnp})$ follows a lognormal distribution with the mean of 1.0 and standard deviation of $\sigma_p$. Additionally, a first-order autoregressive model is employed to represent the temporal correlation in precipitation measurement errors. At each time step, the perturbation is mathematically adjusted as follows:

$$ln\ \delta_n^P(t_i) = \mu_{lnp} + \alpha_{lnp}[ln\ \delta_n^P(t_{i-1}) - \mu_{lnp}]$$
$$+ \varphi\sigma_{lnp}(1 - \alpha_{lnp}^2)^{0.5} \tag{S1-2}$$

Where $\mu_{lnp} = -0.5\sigma_{lnp}^2$ ; $\alpha_{lnp}$ is autocorrelation coefficient for precipitation measurement errors.

**Revised Supplement Part S1.2:**

**S1.2. Uncertainty in observations**

The observation error is generalized as functions of the corresponding observed values (Weerts & El Serafy, 2006; Clark et al., 2008; Alvarez-Garreton et al., 2015):

$$\boldsymbol{y}_j^o(t_i) = [\boldsymbol{I} + \boldsymbol{\delta}^y(t_i)] \cdot \boldsymbol{y}(t_i) \tag{S1-3}$$

Where $\boldsymbol{y}_j^o(t_i) \in \mathcal{R}^{N_y}$ represents the perturbed observation vector for the jth ensemble. $\boldsymbol{I}$ is identity matrix; $\boldsymbol{\delta}^y(t_i)$ is Gaussian perturbation matrix. Assuming that the observation errors are spatially independent, $\boldsymbol{\delta}^y(t_i) \in \mathcal{R}^{N_y \times N_y}$ is a diagonal matrix with diagonal elements $\delta_n^y(t_i), (n = 1, ..., N_y)$. When assimilating soil moisture observations, the diagonal elements follow a normal distribution $\delta_n^y(t_i) \sim N(0, \sigma_{ys})$, and similarly, $\delta_n^y(t_i) \sim N(0, \sigma_{yd})$ is used when assimilating discharge observations. Furthermore, we employ a first-order autoregressive model to account for the temporal correlation in observation errors. At time step $t$, the perturbation is adjusted using the formula:

$$\delta_n^y(t_i) = \mu_y + \alpha_y[\delta_n^y(t_{i-1}) - \mu_y] + \varphi\sigma_y(1 - \alpha_y^2)^{0.5} \tag{S1-4}$$

Where $\mu_y = 0$; $\varphi$ is a standard Gaussian noise; $\sigma_y$ is the standard deviation, which, as previously stated, takes the values $\sigma_{ys}$ or $\sigma_{yd}$ ; $\alpha_y$ is the autocorrelation coefficient, with values of $\alpha_{ys}$ when assimilating soil moisture observations, or $\alpha_{yd}$ when assimilating discharge observations.

4. As far as I know, the Xin'anjiang model is based on the saturation-excess theory, making it suitable only for regions where this runoff generation mechanism dominates, such as humid and semi-humid areas. It is not applicable in regions where infiltration-excess theory is predominant, such as arid and semi-arid areas. Could the authors clarify

whether the method proposed in this study is applicable to arid and semi-arid regions? (page 9, Lines 251-260)

**Response**: Thank you for your discussion of the Xin'anjiang model. We completely agree with your view on its runoff generation mechanism. The Xin'anjiang model is indeed only suitable for humid regions where the saturation-excess runoff mechanism is dominant and is not applicable to arid and semi-arid regions. However, it is important to note that the state updating method proposed in this study is not limited to coupling with the Xin'anjiang model. In fact, this method can be easily coupled with any hydrological model that includes state variables related to soil moisture and channel storage. When coupled with hydrological models suitable for semi-arid and arid regions, it can be effectively applied in those areas. We have discussed this issue in the Discussion section of the revised manuscript (LINES 663-671).

**Revised Manuscript LINES 663-671:**

The Xin'anjiang model is a conceptual hydrological model that generalizes the rainfall-runoff process. Its most prominent feature is performing runoff production calculations based on the saturation-excess runoff mechanism, meaning net rainfall is first entirely used to replenish soil water, and once the soil moisture content in the unsaturated zone reaches field capacity, all subsequent net rainfall is used to generate runoff. Therefore, the Xin'anjiang model is only suitable for humid and semi-humid regions where the saturation-excess runoff mechanism is dominant and is not applicable to arid and semi-arid regions. However, it is important to note that the state updating method proposed in this study is not limited to coupling with the Xin'anjiang model. In fact, this method can be easily coupled with any lumped or semi-distributed hydrological model that includes state variables related to soil moisture and channel storage. When coupled with hydrological models suitable for semi-arid and arid regions, it can be effectively applied in those areas.

5. How are the initial state values for the daily simulation model set? (page 15, Lines 363-364)

**Response**: Thank you for your comment. In this study, the initial values for the daily simulation are set with the soil moisture content at half of the saturation value, and the sub-reaches outflow was set as the observed discharge at the basin outlet on the start date, divided by the total number of sub-reaches. In fact, after an extended period of daily simulation, the initial values of the state variables have a negligible impact on the study, so they can be set to any reasonable value. We have emphasized this point in the revised manuscript (LINES 371-377).

**Revised Manuscript LINES 371-377:**

As long as the warming-up period is adequately long, the influence of initial soil moisture on the simulation at the end of warming-up period, allowing soil moisture for daily simulation to be used as initial conditions for hourly simulation (Yao et al., 2012). The initial values of the daily simulations have a minimal effect on the hourly simulation, so they can be set arbitrarily within reason. In this study, the initial values for the daily simulation are set with the soil moisture content at half of the saturation value, and the sub-reaches outflow is set as the observed discharge at the basin outlet on the start date, divided by the total number of sub-reaches.

**6. Why a longer assimilation time window sometimes leads to poorer results. Could the authors provide an explanation for this? (page 17, Lines 427-433)**

**Response**: Thank you for your comment. This is primarily because a longer time window includes too much historical information, which may have a weak correlation with the current state variables. Including too much historical observational information in the assimilation system may lead to a degradation in assimilation performance. Tao et al. (2016) (https://doi.org/10.1016/j.jhydrol.2016.02.019) tested the performance of the standard AEnKF method with 1-3 hour assimilation time windows and obtained similar results. They found that the 2-hour time window generally yielded better assimilation results than the 3-hour time window, while the 1-hour time window performed the worst. We have discussed this phenomenon in the Discussion section of the revised manuscript (LINES 614-620).

**Revised Manuscript LINES 614-620:**

In the study of assimilation windows for AEnKF in synthetic cases, we found that longer assimilation windows do not necessarily yield better results (Fig. 3). This is primarily because a longer time window includes too much historical information, which may have a weak correlation with the current state variables. Due to the nonlinearity of the hydrological model, where overly long windows can result in the system assimilating excessive noise, which negates the benefits derived from incorporating past observations. Tao et al. (2016) obtained similar results when studying the assimilation window length (1-3 hour) for the assimilation of observed discharge only. They found that the 2-hour time window generally yielded better assimilation results than the 3-hour time window, while the 1-hour time window performed the worst.

**7. What does "One-step prediction" refer to? Does it mean a one-hour forecast? Please clarify. (page 18, Line 444)**

**Response**: We apologize for any confusion caused by this imprecise description. "One-step prediction" indeed refers to a one-hour forecast, and we have clarified this in the revised manuscript (LINE 458).

**Revised Manuscript LINE 458:**

One-step (one-hour) prediction

8. Why was the lead time set to 8 hours? (page 31, Figure 13)

**Response**: Thank you for your comment. In our real-world cases, we selected an 8-hour lead time primarily due to the limitations of data length. To ensure the consistency of the forecast sequence length and the comparability of results, the forecast start time for different lead times within the same flood event was set to the same point -- specifically, the LT hour after the flood start time (the earliest available hourly data). LT represents the longest lead time in this study. If the longest lead time is set to LT = 8 hours, even for a 1-hour lead time, the forecast begins at the 8th hour after the flood start time. Given the overall short length of available hourly data, in some flood events, the peak occurs as early as the 9th or 10th hour after the forecast begins. If the lead time were set longer than 8 hours, the forecast sequence might not include the flood peak, rendering the results meaningless for flood forecasting. Therefore, in the real-data experiments, we set the maximum lead time to 8 hours. To compensate for the shorter lead time in the real-world cases, we extended the maximum lead time to 24 hours in the synthetic data experiments, which is fully adequate for flood forecasting in medium to small basins covering several thousand square kilometers. We have provided additional explanations in the Experimental Setup section of the revised manuscript for synthetic cases (LINES 379-383) and real-world cases (LINES 400-404), respectively.

**Revised Manuscript LINES 379-383:**

In the synthetic cases, the hydrological model operates on an hourly timestep with a maximum lead time of 24 hours, and ensemble simulations involve 100 members. The initial soil moisture is set to half of the maximum value. To ensure consistency in the length of forecast sequences and the comparability of results, the start time for forecasting the same flood event under different lead times is set at the same moment -- specifically, the 24 hours (maximum lead time) after the flood start time.

**Revised Manuscript LINES 400-404:**

In the real-world cases, the timestep and number of ensemble members are the same as in the synthetic cases. Similar to the synthetic cases, to ensure the comparability of results, the forecast start time for all lead times is uniformly delayed from the flood

onset (the earliest available hourly data) by a duration corresponding to the maximum lead time. For some flood events, high flow occurred as early as the 9th hour after onset. To avoid missing the peak flow, the maximum lead time is set to 8 hours.

9. Any limitations of the study should be openly discussed, along with suggestions for future research. (page 32)

**Response**: Thank you for your suggestion. In the revised manuscript, we have included a discussion on the limitations of the methodology used in this study (LINES 662-680).

**Revised Manuscript LINES 662-680:**

**6.3 Limitations**

The Xin'anjiang model is a conceptual hydrological model that generalizes the rainfall-runoff process. Its most prominent feature is performing runoff production calculations based on the saturation-excess runoff mechanism, meaning net rainfall is first entirely used to replenish soil water, and once the soil moisture content in the unsaturated zone reaches field capacity, all subsequent net rainfall is used to generate runoff. Therefore, the Xin'anjiang model is only suitable for humid and semi-humid regions where the saturation-excess runoff mechanism is dominant and is not applicable to arid and semi-arid regions. However, it is important to note that the state updating method proposed in this study is not limited to coupling with the Xin'anjiang model. In fact, this method can be easily coupled with any lumped or semi-distributed hydrological model that includes state variables related to soil moisture and channel storage. When coupled with hydrological models suitable for semi-arid and arid regions, it can be effectively applied in those areas.

Semi-distributed hydrological models, like the Xin'anjiang model used in this study, have smaller state variable dimensions, allowing for the direct application of the proposed state updating scheme. However, in distributed models where each computational grid (e.g., DEM-based grids) has its own state variables, the state dimension becomes large, making direct application inefficient or prone to spurious correlations from distant observations. To resolve this, we recommend applying covariance localization to AEnKF (Janjić et al., 2011) or other localization techniques (Khaniya et al., 2022). For instance, in covariance localization, a localization radius (RL) is set, and the forecast error covariance matrix is adjusted using a correlation matrix derived from the Schur product theorem. This study focuses on jointly assimilating soil moisture and streamflow using AEnKF, and performing localization on AEnKF is beyond the scope of this research. We will explore this further in future

work.

**Special thanks to you for your good comments. Other revisions to the manuscript can be found in " Point-by-point response to Anonymous Reviewer #2" and " Point-by-point response to Zongping Ren's comments".**

**Reference mentioned in the responses**

Tao, J., Wu, D., Gourley, J., Zhang, S. Q., Crow, W., Peters-Lidard, C., & Barros, A. P.: Operational hydrological forecasting during the IPHEx-IOP campaign - Meet the challenge. J. Hydrol., 541, 434-456, https://doi.org/10.1016/j.jhydrol.2016.02.019, 2016.

**Point-by-point response to Anonymous Reviewer #2**

We would like to sincerely appreciate you for the review of our manuscript "State updating in the Xin'anjiang Model: Joint assimilating streamflow and multi-source soil moisture data via Asynchronous Ensemble Kalman Filter with enhanced Error Models" and the constructive suggestions. We sincerely believe these comments facilitate the quality improvement of this manuscript. All the comments have been considered and a point-by-point response has been provided below.

The point-by-point response is formatted as follows:
- Reviewer's comments are shown in blue
- Authors' response are shown in black
- Authors' changes in the manuscript are shown in red. The line numbers indicated in this response are those in the "Revised Manuscript with no changes marked" document
- The unchanged parts of the manuscript are shown in black
* * *
In the manuscript, joint assimilating streamflow and soil moisture data via Asynchronous Ensemble Kalman Filter with enhanced Error Models was conducted. The modelling results are improved compared with conventional methods. The findings are very helpful for real-time flood forecast. The following points should be further clarified in the revised version.

**Response**: Thank you for your concise summary of the paper and for your positive feedback on our study. We have carefully considered all of your comments and responded them in the subsequent specific comments section.

**Specific Comments:**
============

1. Methods section, I suggest 'hydrological model' should be introduced first. Then the readers could understand the model parameters easily in other sections.

**Response:** Thank you for your suggestion. We have adjusted the structure of the Methodology and method section, beginning with an introduction to the hydrological model (LINES 164-189).

**Revised Manuscript LINES 164-189:**

**2 Methodology and method**

**2.1 Hydrological model**

The Xin'anjiang model, conceptualized by Zhao (1992), is a distinguished hydrological model, primarily based on a saturation excess mechanism. Renowned for its straightforward structure and explicit parameter definitions, this model excels in simulating humid catchments, making it a popular tool for flood forecasting in in China. To account for spatial variability in rainfall distribution and surface characteristics, the model typically segments a catchment into several sub-basins. These sub-basins act as computational units for runoff generation and routing.

The Xin'anjiang model demands relatively straightforward driving data, and key inputs include the areal mean rainfall depth (P) and pan evaporation (EM) for each sub-basin. The model typically comprises four main components: evapotranspiration, runoff production, runoff separation, and flow routing, involving the calibration of 16 distinct parameters. The flow chart of the Xin'anjiang model is presented in Fig. 1. Soil evaporation is derived from pan evaporation data using a 'three-layer soil moisture module'. The runoff generation is based on a saturation-excess mechanism, where runoff is produced only when the soil moisture in the unsaturated zone reaches field capacity. The 'lag and route' method calculates the outflow from each sub-basin. Flow routing from the sub-basin outlets to the total basin outlet employs the Muskingum method to successive sub-reaches. It is implemented through dividing the channel from each sub-basin outlet to the total basin outlet into varying numbers of sub-reaches. These sub-reaches are based on the distance from each sub-basin outlet to the total basin outlet. In addition, the basin inflow is directly calculated to the outlet by the Muskingum method.

[Figure]

**Fig. 1. Flow Chart of Xin'anjiang Model. The variables in the boxes indicate the model state, inputs and outputs, and the symbols outside the corresponding blocks are model parameters.**

Zhao (1992) categorized the parameters of Xin'anjiang model into sensitive and non-sensitive groups. In real-world cases, non-sensitive parameters are assigned values based on expert judgment, while optimal values for sensitive parameters are derived from historical data using the Shuffled Complex Evolution (SCE-UA) method (Duan et al., 1992). For synthetic cases, however, parameters are taken as recommend defaults. Table 1 summarizes these parameters.

**Table 1. Parameters of the Xin'anjiang model**

| Parameter [a] | Description | Synthetic cases | Real-world cases |
|---|---|---|---|
| ***K*** | the ratio of potential evapotranspiration to pan evaporation | 1.00 | 0.95 |
| *C* | Evapotranspiration coefficient of deeper layer | 0.13 | 0.05 |
| *WUM* | Averaged tension water capacity of upper layer (mm) | 12.5 | 19.9 |
| *WLM* | Averaged tension water capacity of lower layer (mm) | 75.0 | 64.4 |
| *B* | Exponent of the tension water capacity curve | 0.40 | 0.38 |
| *WM* | Averaged tension water capacity (mm) | 125.0 | 119.8 |
| *IM* | Percentage of impervious areas in the catchment | 0.01 | 0.03 |
| ***SM*** | Averaged free water storage capacity (mm) | 30.0 | 16.7 |
| *EX* | Exponent of the free water capacity curve | 1.25 | 1.50 |
| ***KI*** | Daily outflow coefficient of free water storage to interflow | 0.35 | 0.02 |
| ***KG*** | Daily outflow coefficient of free water storage to groundwater | 0.35 | 0.68 |
| *CI* | Daily recession constant of the interflow storage | 0.70 | 0.52 |
| ***CG*** | Daily recession constant of the groundwater storage | 0.99 | 0.93 |
| ***CS*** | Daily recession constants of channel network storage | 0.50 | 0.88 |
| ***LAG*** | Lag in time (h) | 0 | 1 |
| *XE* | Parameters of the Muskingum method | 0.25 | 0.01 |

[a] Parameters in bold and underline text indicate sensitive parameters.

2. Figure 2(b), there are 3 discharge stations, namely Hexi, Gaochetou, and Wuqiangxibashang. But it is hard to see the controlled drainage area for these 3 stations. Although the rainfall station, soil moisture monitoring sites are can be seen, it should be described in the main text.

**Response**: We apologize for the misunderstanding caused by the unclear image description. To clarify, of the three hydrological stations, Wuqiangxibashang provides outflow data at the basin outlet, while Hexi and Gaochetou provide inflow data to the basin. However, due to the lack of soil moisture and rainfall data within their control areas, the control areas of Hexi and Gaochetou station were not included in this study. We have emphasized this point in the "Study areas and data" section of the revised manuscript (LINES 356-359).

**Revised Manuscript LINES 356-359:**

Among the three discharge stations in the study catchment, Wuqiangxibashang provides the outflow data at the outlet, while Hexi and Gaochetou are stations that provide inflow data for the study area. Due to the lack of soil moisture and rainfall data within their controlled areas, the control areas of Hexi and Gaochetou are not included in the study. For an overview of the data used in this study, please see Supplement 4.

3. In the study region, is there any hydraulic infrastructure to affect runoff generation?

**Response**: Thank you for your common. The study area is a natural watershed, and the only nearby large reservoir is located downstream of the Wuqiangxibashang station. As a result, it does not significantly impact the forecast results for the study area.

4. Line 389, 'the maximum lead time is set to 8 hours to avoid missing peak flows'. I cannot understand the linkage between lead time and peak flows.

**Response**: Thank you for your comment. This issue was also addressed in our response to Anonymous Reviewer #1. In our real-world cases, we selected an 8-hour lead time primarily due to the limitations of data length. To ensure the consistency of the forecast sequence length and the comparability of results, the forecast start time for different lead times within the same flood event was set to the same point -- specifically, the LT hour after the flood start time (the earliest available hourly data). LT represents the longest lead time in this study. If the longest lead time is set to LT = 8 hours, even for a 1-hour lead time, the forecast begins at the 8th hour after the flood start time. Given the overall short length of available hourly data, in some flood events, the peak occurs as early as the 9th or 10th hour after the forecast begins. If the lead time were set longer than 8 hours, the forecast sequence might not include the flood peak, rendering the results meaningless for flood forecasting. Therefore, in the real-data experiments, we set the maximum lead time to 8 hours. To compensate for the shorter lead time in the real-world cases, we extended the maximum lead time to 24 hours in the synthetic data experiments, which is fully adequate for flood forecasting in medium to small basins covering several thousand square kilometers. We have provided additional explanations in the Experimental Setup section of the revised manuscript for synthetic cases (LINES 379-383) and real-world cases (LINES 400-404), respectively.

**Revised Manuscript LINES 379-383:**

In the synthetic cases, the hydrological model operates on an hourly timestep with a maximum lead time of 24 hours, and ensemble simulations involve 100 members. The initial soil moisture is set to half of the maximum value. To ensure consistency in the

length of forecast sequences and the comparability of results, the start time for forecasting the same flood event under different lead times is set at the same moment -- specifically, the 24 hours (maximum lead time) after the flood start time.

**Revised Manuscript LINES 400-404:**

In the real-world cases, the timestep and number of ensemble members are the same as in the synthetic cases. Similar to the synthetic cases, to ensure the comparability of results, the forecast start time for all lead times is uniformly delayed from the flood onset (the earliest available hourly data) by a duration corresponding to the maximum lead time. For some flood events, high flow occurred as early as the 9th hour after onset. To avoid missing the peak flow, the maximum lead time is set to 8 hours.

5. Discussion is an important part. I suggest it be a separate section. If the proposed method are used in distributed hydrological models (i.e. distributed Xin'anjiang model), what will be the results?

**Response**: Thank you for your suggestion, which has been extremely helpful for improving the paper. In the revised manuscript, we have included a separate discussion section, focusing on topics such as typical flood events and the limitations of the proposed method (LINES 612-680). This includes the challenges of applying the method to distributed hydrological models. Semi-distributed hydrological models, like the Xin'anjiang model used in this study, have smaller state variable dimensions, allowing for the direct application of the proposed state updating scheme. However, in distributed models where each computational grid (e.g., DEM-based grids) has its own state variables, the state dimension becomes large, making direct application inefficient or prone to spurious correlations from distant observations. To resolve this, we recommend applying covariance localization to AEnKF (Janjić et al., 2011, https://doi.org/10.1175/2011MWR3552.1) or other localization techniques (Khaniya et al., 2022, https://doi.org/10.1016/j.jhydrol.2022.127651). For instance, in covariance localization, a localization radius $R_L$ is set, and the forecast error covariance matrix is adjusted using a correlation matrix derived from the Schur product theorem. This study focuses on jointly assimilating soil moisture and streamflow using AEnKF, and performing localization on AEnKF is beyond the scope of this research. We will explore this further in future work.

**Revised Manuscript LINES 612-680:**

**6. Discussion**

**6.1 Discussion of AEnKF time window in synthetic cases**

[revised manuscript text omitted]

Semi-distributed hydrological models, like the Xin'anjiang model used in this study, have smaller state variable dimensions, allowing for the direct application of the proposed state updating scheme. However, in distributed models where each computational grid (e.g., DEM-based grids) has its own state variables, the state dimension becomes large, making direct application inefficient or prone to spurious correlations from distant observations. To resolve this, we recommend applying covariance localization to AEnKF (Janjić et al., 2011) or other localization techniques (Khaniya et al., 2022). For instance, in covariance localization, a localization radius (RL) is set, and the forecast error covariance matrix is adjusted using a correlation matrix derived from the Schur product theorem. This study focuses on jointly assimilating soil moisture and streamflow using AEnKF, and performing localization on AEnKF is beyond the scope of this research. We will explore this further in future

work.

6. Section 5.1.3, only 6 flood events are selected for analysis, could you add some flood events in 2024? Could you please provide the simulated hydrographs by the assimilation schemes for the 6 events?

**Response**: Thank you for your suggestion. We have realized that the most recent year's flood data was not utilized. Recently, we collected data of two flood events (No.2024040100 and No.2024042900) in 2024, as shown in Section S4 of the supplement document (LINES S106-S123). We have added simulations and analyses of these two flood events in both the synthetic and real-world cases. Consequently, Figures 3-11, Table 3, and the corresponding results have been updated. The new results can be found in "5 Results" section of the revised manuscript (LINES 416-611) and are not presented here. It is important to emphasize that the addition of the 2024 flood events did not alter the main conclusions of this study, which potentially further validates the general applicability of the proposed method.

We have also included the hydrographs for all eight flood events, please see the revised supplement document (S5. Hydrographs in Real-world Cases).

**Revised Supplement LINES S106-S123:**

This hydro-meteorological data utilized in the study spanning from 2014 to 2024, provided by the Hunan Provincial Hydrological Bureau, including evaporation, precipitation, and discharge data. Within the catchment, there are 17 rain gauges, one evaporation observation station, and four discharge observation stations. Evaporation data are derived from daily pan evaporation measurements using the E-601 pan, with hourly values calculated as 1/24th of the daily measurements. Notably, with only one evaporation observation station in the catchment, it is assumed that the observed evaporation is spatially uniform. When multiple rain gauges exist within a sub-catchment, the area-averaged rainfall is calculated as the arithmetic mean of all gauge observations. For discharge observation stations, Wuqiangxibashang (WQXBS) serves as the outlet observation station, while the remaining three stations Hexi (HX), Pushi (PS), and Gaochetou (GCT) measure inflow. Hourly observations of precipitation and discharge are intermittent, thus hourly data are only available during flood events, with daily data available at other times. Fifteen flood events from 2014 to 2018 were used for model calibration, and sixteen events from 2019 to 2024 for model validation. Considering soil moisture data availability, eight flood events in 2023 and 2024 were used for assimilation studies. For an overview of these flood events, refer to Table S4-1. The statistical characteristics of the observed and simulated peak flows are presented in Table S4-2.

**Table S4-1. List of flood events investigated in this study**

| | Serial number | Start date | End date | Observed Peak flow (m³/s) | Simulated Peak flow (m³/s) |
|---|---|---|---|---|---|
| calibration | No.2014052300 | 2014/05/23 00:00 | 2014/05/27 20:00 | 17356 | 17335 |
| | No.2014070300 | 2014/07/03 00:00 | 2014/07/06 08:00 | 22705 | 21564 |
| | No.2014071400 | 2014/07/14 00:00 | 2014/07/19 00:00 | 35725 | 35648 |
| | No.2015060121 | 2015/06/01 21:00 | 2015/06/07 01:00 | 17762 | 17085 |
| | No.2015060718 | 2015/06/07 18:00 | 2015/06/10 18:00 | 12017 | 11018 |
| | No.2015062023 | 2015/06/20 23:00 | 2015/06/24 09:00 | 19196 | 16971 |
| | No.2016050703 | 2016/05/07 03:00 | 2016/05/11 06:00 | 13051 | 12191 |
| | No.2016062017 | 2016/06/20 17:00 | 2016/06/21 21:00 | 12472 | 10268 |
| | No.2016062720 | 2016/06/27 20:00 | 2016/06/30 03:00 | 14996 | 13072 |
| | No.2016070311 | 2016/07/03 11:00 | 2016/07/08 12:00 | 22278 | 21016 |
| | No.2017052208 | 2017/05/22 08:00 | 2017/05/25 19:00 | 8872 | 8926 |
| | No.2017062711 | 2017/06/27 11:00 | 2017/07/05 12:00 | 32147 | 32121 |
| | No.2017081121 | 2017/08/11 21:00 | 2017/08/16 00:00 | 13091 | 14958 |
| | No.2018053010 | 2018/05/30 10:00 | 2018/06/03 16:00 | 7348 | 7462 |
| | No.2018092518 | 2018/09/25 18:00 | 2018/09/27 05:00 | 8518 | 7495 |
| validation | No.2019051905 | 2019/05/19 05:00 | 2019/05/22 00:00 | 14024 | 13142 |
| | No.2019070700 | 2019/07/07 00:00 | 2019/07/16 12:00 | 14046 | 13358 |
| | No.2020070800 | 2020/07/08 00:00 | 2020/07/09 18:00 | 25963 | 23428 |
| | No.2020071823 | 2020/07/18 23:00 | 2020/07/20 16:00 | 18688 | 15459 |
| | No.2020091500 | 2020/09/15 00:00 | 2020/09/21 08:00 | 20829 | 20393 |
| | No.2021050300 | 2021/05/03 00:00 | 2021/05/05 00:00 | 8021 | 8397 |
| | No.2021051112 | 2021/05/11 12:00 | 2021/05/27 00:00 | 13347 | 12433 |
| | No.2021060300 | 2021/06/03 00:00 | 2021/06/07 00:00 | 8391 | 7693 |
| | **No.2023040308** | 2023/04/03 08:00 | 2023/04/05 14:00 | 6192 | 7891 |
| | **No.2023050416** | 2023/05/04 16:00 | 2023/05/06 17:00 | 4747 | 4244 |
| | **No.2023052008** | 2023/05/20 08:00 | 2023/05/22 18:00 | 5660 | 7702 |
| | **No.2023062100** | 2023/06/21 00:00 | 2023/06/25 19:00 | 6940 | 5834 |
| | **No.2023063000** | 2023/06/30 00:00 | 2023/07/01 14:00 | 9317 | 7809 |
| | **No.2023072516** | 2023/07/25 16:00 | 2023/07/27 18:00 | 8449 | 7611 |
| | **No.2024040100** | 2024/04/01 00:00 | 2024/04/03 01:00 | 5430 | 6286 |
| | **No.2024042900** | 2023/04/29 00:00 | 2024/05/01 17:00 | 5735 | 5754 |

[a] The flood events utilized for assimilation research are indicated by bold text with an underline.

**Table S4-2. Statistical characterization of peak flow**

| | | | Mean (m³/s) | standard deviation (m³/s) | Minim-um (m³/s) | Maxi-mum (m³/s) | Median (m³/s) | Skew-ness | Kurt-osis | Coefficient of Variation | 95% confidence interval (m3/s) |
|---|---|---|---|---|---|---|---|---|---|---|---|
| observed peak flow | | Calibr-ation | 15982 | 8049 | 7348 | 35725 | 14996 | 1.03 | 0.55 | 0.50 | (11534, 20431) |
| | | Valid-ation | 11255 | 6343 | 4747 | 25963 | 8420 | 0.89 | 0.37 | 0.56 | (7879, 14630) |
| Simulated Peak flow | | Calibr-ation | 15942 | 8142 | 7462 | 35648 | 14958 | 0.97 | 0.22 | 0.51 | (11444, 20440) |
| | | Valid-ation | 10596 | 5669 | 4244 | 23428 | 7850 | 0.87 | -0.15 | 0.53 | (7578, 13614) |

**Revised Supplement Part S5:**
**S5. Hydrographs in Real-world Cases**

[Figure]

[Figure]

**Figure S5-1. Hydrographs in real-world cases. The left panel shows the AEnKF$_Q$ scheme, the center panel shows the AEnKF$_S$ scheme, and the right panel shows the AEnKF$_{SQ}$ scheme**

Special thanks to you for your good comments. Other revisions to the manuscript can be found in " Point-by-point response to Anonymous Reviewer #1" and " Point-by-point response to Zongping Ren's comments".

**Reference mentioned in the responses**

Janjić, T., Nerger, L., Albertella, A., Schröter, J., and Skachko, S.: On domain localization in ensemble-based Kalman filter algorithms. Mon. Weather Rev., 139(7), 2046-2060, https://doi.org/10.1175/2011MWR3552.1, 2011.

Khaniya, M., Tachikawa, Y., Ichikawa, Y., and Yorozu, K.: Impact of assimilating dam outflow measurements to update distributed hydrological model states: Localization for improving ensemble Kalman filter performance. J. Hydrol., 608, 127651, https://doi.org/10.1016/j.jhydrol.2022.127651, 2022.

**Point-by-point response to Zongping Ren's comments**

We would like to sincerely appreciate you for the review of our manuscript "State updating in the Xin'anjiang Model: Joint assimilating streamflow and multi-source soil moisture data via Asynchronous Ensemble Kalman Filter with enhanced Error Models" and the constructive suggestions. We sincerely believe these comments facilitate the quality improvement of this manuscript. All the comments have been considered and a point-by-point response has been provided below.

The point-by-point response is formatted as follows:
- Reviewer's comments are shown in blue
- Authors' response are shown in black
- Authors' changes in the manuscript are shown in red. The line numbers indicated in this response are those in the "Revised Manuscript with no changes marked" document
- The unchanged parts of the manuscript are shown in black
* * *
The study is briefly based on the development of the Xin'anjiang hydrological model. For this aim, Asynchronous Ensemble Kalman Filter (AEnKF) with enhanced error model is used to joint assimilate streamflow and multi-source soil moisture data. Furthermore, this paper proposes a novel method to integrate CLDAS soil moisture data with in situ observations, enhancing the accuracy of the dataset. Wuqiangxi catchment is selected for the application. The results produced by the AEnKF assimilating different types of observations are then evaluated by some performance metrics. The work is extensive and well-structured. The subject is novel and the study is valuable in terms of the hydrological forecasting in terms of flood events in river basins. However, the discussion of main and latest studies on the subject needs to be further strengthened. Some suggestions and comments to the authors are presented below:

**Response**: Thank you for your concise summary of the paper and for your positive feedback on our study. We have carefully considered all of your comments and responded them in the subsequent specific comments section.

**Specific Comments:**

============

1. What are main differences between AEnKF and EnKF? Supported and related studies about AEnKF should be strongly presented in Introduction to emphasize highlights of the paper.

**Response**: Thank you for your comment. The EnKF is a synchronous assimilation method that assimilates observations at the current time into the hydrological model at the analysis step. This means that EnKF updates the state variables of the Xin'anjiang model based only on observations from the current time step. In contrast, AEnKF is a more advanced asynchronous assimilation method, allowing for the assimilation of both current and past observations during the analysis step. Specifically, in this study, AEnKF assimilates observations from the current time and the previous $t_w$ hours ($t_w$ is the assimilation time window) into the Xin'anjiang model, updating the model's state variables. This asynchronous assimilation helps to consider the complex nonlinear relationships between observations at multiple times and the hydrological model's state variables. We have included a discussion on existing research and applications of the AEnKF method in the Introduction section of the revised manuscript (LINES 77-90).

**Revised Manuscript LINES 77-90:**

The AEnKF technique was first applied by Krymskaya (2013) to the problem of history matching in reservoir engineering. The study revealed that AEnKF outperforms EnKF in parameter estimation and utilizes the data with similar efficiency. The AEnKF is recognized for its simplicity and high computational efficiency, offering significant potential in short-term flood forecasting applications. Despite its promise, the scope of research in this area is relatively limited. Among the few studies conducted, Mazzoleni et al. (2018) evaluated AEnKF assimilation in simplified flow routing models, highlighting its exceptional performance in both lumped and distributed flow routing. Tao et al. (2016) summarized the hydrological forecasting test conducted during the 2014 IPHEx-IOP campaign, proposing a framework for improving flood prediction in mountainous regions through the assimilation of discharge data using the AEnKF method, with a focus on enhancing forecast accuracy and reducing uncertainty. In addition, Rakovec et al. (2015) and our earlier study (Gong et al., 2024) applied the AEnKF to the distributed HBV-96 model and the Xin'anjiang model, respectively. These studies examined effectiveness of AEnKF in real-time correction through the assimilation of observed discharge in distributed and semi-distributed hydrological models, revealing that AEnKF outperforms the standard EnKF. However, these studies assimilate only a single type of observational data (e.g., observed discharge) using the AEnKF method, which does not take full advantage of the AEnKF.

2. Line 231. The SM is mentioned but not explained. Please insert a definition the first time it is mentioned.

**Response**: Thank you for your suggestion. In response to the suggestions from Anonymous Reviewer #2, we have revised the structure of the paper by introducing the hydrological model at the beginning of the "Methodology and method" section. As a result, the definition of the parameter SM, averaged free water storage capacity, is now provided the first time it is mentioned in Table 1.

**Revised Manuscript Table 1:**

**Table 1. Parameters of the Xin'anjiang model**

| Parameter [a] | Description | Synthetic cases | Real-world cases |
|---|---|---|---|
| **K** | the ratio of potential evapotranspiration to pan evaporation | 1.00 | 0.95 |
| *C* | Evapotranspiration coefficient of deeper layer | 0.13 | 0.05 |
| *WUM* | Averaged tension water capacity of upper layer (mm) | 12.5 | 19.9 |
| *WLM* | Averaged tension water capacity of lower layer (mm) | 75.0 | 64.4 |
| *B* | Exponent of the tension water capacity curve | 0.40 | 0.38 |
| *WM* | Averaged tension water capacity (mm) | 125.0 | 119.8 |
| *IM* | Percentage of impervious areas in the catchment | 0.01 | 0.03 |
| **SM** | Averaged free water storage capacity (mm) | 30.0 | 16.7 |
| *EX* | Exponent of the free water capacity curve | 1.25 | 1.50 |
| **KI** | Daily outflow coefficient of free water storage to interflow | 0.35 | 0.02 |
| **KG** | Daily outflow coefficient of free water storage to groundwater | 0.35 | 0.68 |
| *CI* | Daily recession constant of the interflow storage | 0.70 | 0.52 |
| **CG** | Daily recession constant of the groundwater storage | 0.99 | 0.93 |
| **CS** | Daily recession constants of channel network storage | 0.50 | 0.88 |
| **LAG** | Lag in time (h) | 0 | 1 |
| *XE* | Parameters of the Muskingum method | 0.25 | 0.01 |

[a] Parameters in bold and underline text indicate sensitive parameters.

3. Section 2.3. I am not clear on how the hydrological model simulates infiltration. What are the infiltration parameters? They don't seem to be shown in Table 1. It is not necessary to explain your hydrological model again in the manuscript, but I need to understand why infiltration parameters are not considered in the model perturbation.

**Response**: Thank you for your comment. We have addressed your questions individually below.

First, we should explain to you the method of calculating the runoff of the Xin'anjiang model. The Xin'anjiang model is a conceptual hydrological model that generalizes the rainfall-runoff process. Its most prominent feature is performing runoff production calculations based on the saturation-excess runoff mechanism, meaning net

rainfall is first entirely used to replenish soil water, and once the soil moisture content in the unsaturated zone reaches field capacity, all subsequent net rainfall is used to generate runoff. Therefore, the Xin'anjiang model does not involve infiltration parameters. In the revised manuscript, we have explained the saturation-excess runoff mechanism in the "Hydrological Model" section (LINES 175-176). Detailed theoretical derivations of the soil evapotranspiration and runoff generation in the Xin'anjiang model are provided below. If you are interested in other aspects of the Xin'anjiang model, we recommend the famous paper "The Xinanjiang Model Applied in China" (Zhao, 1992, https://doi.org/10.1016/0022-1694(92)90096-E).

**(1) Evapotranspiration**

The Xin'anjiang model divides the soil into upper, lower, and deep layers based on vertical heterogeneity. It uses a three-layer evapotranspiration model to calculate actual evapotranspiration, involving parameters such as upper layer tension water capacity ($WUM$, mm), lower layer tension water capacity ($WLM$, mm), average basin tension water capacity ($WM$, mm), evapotranspiration conversion coefficient ($K$), and deep layer evapotranspiration coefficient ($C$). The three-layer evapotranspiration model is with the following principles: the upper layer evaporates according to its evapotranspiration capacity; if the upper layer's soil moisture content is insufficient, the remaining evapotranspiration capacity is drawn from the lower layer. The lower layer's evaporation is proportional to the evapotranspiration capacity and its soil moisture storage, with the ratio of the calculated lower layer evaporation to the remaining evapotranspiration capacity not less than the deep layer evapotranspiration coefficient. If this ratio is not met, the deficit is supplemented by the lower layer water storage. If the lower layer water storage is insufficient, it is supplemented by the deep layer water storage. The calculation formula for the three-layer evapotranspiration model can be summarized as follows.

First, the evapotranspiration capacity $EP$ (mm) is calculated using the pan evaporation ($EM$).

$$EP = K \cdot EM \tag{R1}$$

When $P + WU \geq EP$, the evapotranspiration for the upper, lower, and deep layers ($EU$, $EL$, and $ED$) are:

$$\begin{cases} EU = EP \\ EL = 0 \\ ED = 0 \end{cases} \tag{R2}$$

When $P + WU < EP$, the upper layer evapotranspiration ($EU$) is:

$$EU = P + WU \tag{R3}$$

On this basis, the calculation of the lower layer evapotranspiration ($EL$) and the deep layer evapotranspiration ($ED$) is divided into three cases:

1) When $WL \geq C \cdot LM$,

$$\begin{cases} EL = (EP - EU) \cdot WL/WLM \\ ED = 0 \end{cases} \tag{R4}$$

2) When $WL < C \cdot LM$ and $WL \geq C \cdot (EP - EU)$,

$$\begin{cases} EL = C \cdot (EP - EU) \\ ED = 0 \end{cases} \tag{R5}$$

3) When $WL < C \cdot LM$ and $WL < C \cdot (EP - EU)$,

$$\begin{cases} EL = WL \\ ED = C \cdot (EP - EU) - WL \end{cases} \tag{R6}$$

Once the upper, lower, and deep layer evapotranspiration ($EU$, $EL$, and $ED$) are fully calculated, the total evapotranspiration $E$ (mm) is the sum of these three amounts:

$$E = EU + EL + ED \tag{R7}$$

**(2) Runoff generation**

The Xin'anjiang model uses the saturation-excess runoff generation method to calculate runoff. Net rainfall ($P$-$E$) first replenishes soil moisture, with no runoff generated until the soil moisture reaches field capacity. Once the soil is saturated, all net rainfall contributes to runoff. The Xin'anjiang model uses a tension water capacity curve to characterize the spatial heterogeneity of soil moisture in the catchment, represented as:

$$\frac{f_A}{Area} = \left[ 1 - \left( 1 - \frac{W'}{WMM} \right)^B \right] (1 - IM) + IM \tag{R8}$$

Where, $f_A$ is the runoff production area (km$^2$); $Area$ is the basin area (km$^2$); $W'$ is the point tension water capacity in the basin (mm); $WMM$ is the maximum point tension water capacity (mm); $WM$ is the average basin tension water capacity (mm); $B$ is the exponent of the tension water storage capacity curve; $IM$ is the proportion of the impermeable area to the total basin area. Let $W$ be the average basin tension water storage at the current time (mm), and $\xi_W$ be the vertical coordinate of $W$ on the tension water capacity curve (mm). Integrating $W'$ from 0 to $\xi_W$ in Eq. (R8) yields:

$$W = \frac{(1 - IM) \cdot WMM}{B + 1} \left[ 1 - \left( 1 - \frac{\xi_W}{WMM} \right)^{B+1} \right] \tag{R9}$$

Substituting $\xi_W = WMM$ and $W = WM$ into Eq. (R9) yields:

$$WMM = \frac{WM \cdot (B + 1)}{(1 - IM)} \tag{R10}$$

Substituting Eq. (R10) into Eq. (R9) yields:

$$\xi_W = WMM \left[ 1 - \left( 1 - \frac{W}{WM} \right)^{\frac{1}{1+B}} \right] \qquad \text{(R11)}$$

The total runoff $R$ (expressed in runoff depth, mm) can be expressed as:

$$R = \int_{\xi_W}^{P-E+\xi_W} \frac{f_A}{Area} dW' \qquad \text{(R12)}$$

No runoff is generated when $P - E \le 0$. When $P - E > 0$, runoff is generated, and the total runoff $R$ is calculated as follows in two scenarios:

1) When $P - E + \xi_W < WMM$,

$$R = P - E - WM + W + WM \left[ 1 - \left( \frac{P - E + \xi_W}{WMM} \right)^{(1+B)} \right] \qquad \text{(R13)}$$

2) When $P - E + \xi_W \ge WMM$,

$$R = P - E - WM + W \qquad \text{(R14)}$$

The Xin'anjiang model considers the vertical regulation of the vadose zone and uses a free water storage reservoir to divide the total runoff $R$ into surface runoff $RS$, interflow $RI$, and groundwater runoff $RG$. The parameters involved include the averaged free water storage capacity $SM$, the exponent of the free water capacity curve $EX$, the daily outflow coefficient of free water storage to groundwater $KG$, and the daily outflow coefficient of free water storage to interflow $KI$.

Considering that the free water storage capacity is also spatially heterogeneous, the Xin'anjiang model uses the free water capacity curve to represent this heterogeneity:

$$\frac{f_A}{Area} = \left[ 1 - \left( 1 - \frac{S'}{SMM} \right)^B \right] \qquad \text{(R15)}$$

Using a derivation similar to that of the tension water capacity curve, we obtain:

$$SMM = SM(EX + 1) \qquad \text{(R16)}$$

$$\xi_S = SMM \left[ 1 - \left( 1 - \frac{S}{SM} \right)^{\frac{1}{1+EX}} \right] \qquad \text{(R17)}$$

where $S'$ is the point free water storage capacity in the basin (mm); $SMM$ is the maximum point free water storage capacity (mm); $SM$ is the average free water storage capacity (mm); $EX$ is the exponent of the free water strage capacity curve; $S$ is the average free water storage at the calculation timestep (mm); and $\xi_S$ is the vertical coordinate of $S$ on the free water capacity curve (mm).

The calculation for surface runoff $RS$ is divided into two cases:

1) When $P - E + \xi_S < SMM$,

$$RS = \left\{ P - E - SM + S + SM \left[ 1 - \left( \frac{P - E + \xi_S}{SMM} \right)^{(1+EX)} \right] \right\} FR \qquad \text{(R18)}$$

2) When $P - E + \xi_S \geq SMM$,

$$RS = (P - E - SM + S) \cdot FR \qquad \text{(R19)}$$

The corresponding interflow $RI$ and groundwater runoff $RG$ are:

$$RI = \left( S + \frac{R - RS}{FR} \right) \cdot KI \cdot FR \qquad \text{(R20)}$$

$$RG = \left( S + \frac{R - RS}{FR} \right) \cdot KG \cdot FR \qquad \text{(R21)}$$

**Revised Manuscript LINES 175-176:**

The runoff generation is based on a saturation-excess mechanism, where runoff is produced only when the soil moisture in the unsaturated zone reaches field capacity.

4. Fig. 2. I suggest replacing the Yangtze River Basin with China to make it more understandable to the reader.

**Response**: Thank you for your suggestion. We have improved Figure 2 in the revised manuscript according to your suggestion.

**Revised Manuscript Figure 2:**

[Figure]

**Fig. 2 Study catchment. (a) Digital Elevation Map (DEM); (b)Sub-basins and observation stations; (c) Soil texture (0 to 30 cm); (d) Soil texture (30 to 100 cm).**

5. Section 5.1.2, Fig. 3. Could the authors explain why sometimes larger time window produces poorer results instead? Is it because observations that go too far back in time compromise the quality of real time?

**Response**: Thank you for your comment. As in the response to Anonymous Reviewer #1, this is primarily because a longer time window includes too much historical information, which may have a weak correlation with the current state variables. Including too much historical observational information in the assimilation system may lead to a degradation in assimilation performance. Tao et al. (2016) (https://doi.org/10.1016/j.jhydrol.2016.02.019) tested the performance of the standard AEnKF method with 1-3 hour assimilation time windows and obtained similar results. They found that the 2-hour time window generally yielded better assimilation results than the 3-hour time window, while the 1-hour time window performed the worst. We have discussed this phenomenon in the Discussion section of the revised manuscript (LINES 614-620).

**Revised Manuscript LINES 614-620:**

In the study of assimilation windows for AEnKF in synthetic cases, we found that longer assimilation windows do not necessarily yield better results (Fig. 3). This is primarily because a longer time window includes too much historical information, which may have a weak correlation with the current state variables. Due to the nonlinearity of the hydrological model, where overly long windows can result in the system assimilating excessive noise, which negates the benefits derived from incorporating past observations. Tao et al. (2016) obtained similar results when studying the assimilation window length (1-3 hour) for the assimilation of observed discharge only. They found that the 2-hour time window generally yielded better assimilation results than the 3-hour time window, while the 1-hour time window performed the worst.

6. Line 505-520, Fig. 7. There are flood events where the performance of the AEnKF remains superior to the OL even after 24 hours. Have authors provided an explanation for why the model correction persists for such an extended period in these cases?

**Response**: Thank you for your comment. We believe that the assimilation effect of AEnKF can last for more than 24 hours, mainly because the soil moisture state variables were effectively updated in these events. The initial soil moisture state at the forecast start time reflects the basin's wetness at that moment and significantly impacts forecast accuracy for a considerably long lead time. In Rakovec et al. (2015) (https://doi.org/10.5194/hess-19-2911-2015), the average temporal persistence of the

standard AEnKF assimilation effect could reach a 45-hour lead time, likely because they also updated the soil moisture state variables.

7. Section 6. This part needs some improvements. Currently, it is presented in broad terms. Specifically, the conclusions drawn from the calculated performance metrics should be detailed explicitly.

**Response**: Thank you for your suggestion. We have added the results of specific performance metrics to the Conclusion section of the revised manuscript (LINES 698-719), making the conclusions more objective.

**Revised Manuscript LINES 698-719:**

In synthetic case studies, while updating soil moisture state variables of the Xin'anjiang model, it is observed that effective updates are limited to free water storage and total tension water storage. This underscores the significance of choosing appropriate state variables for updates in the application of the AEnKF method. Further analysis revealed that with high-quality, hourly available observational data, all three assimilation schemes maintained their effectiveness for up to 24-hour lead time. Notably, $\text{AEnKF}_{SQ}$ demonstrated enhanced optimal single-value performance, overall ensemble performance, and ensemble reliability, surpassing both $\text{AEnKF}_S$ and $\text{AEnKF}_Q$. Specifically, in the one-step forecast, the $\text{MR}_{RMSE}$ for $\text{AEnKF}_{SQ}$ decreased by 0.11 and 0.16 compared to $\text{AEnKF}_S$ and $\text{AEnKF}_Q$, respectively; the $\text{MR}_{CRPS}$ for $\text{AEnKF}_{SQ}$ decreased by 0.10 and 0.15, and the $\text{MR}_{RELI}$ decreased by 0.20 and 0.15 compared to $\text{AEnKF}_S$ and $\text{AEnKF}_Q$, respectively. $\text{AEnKF}_{SQ}$'s advantage in optimal single-value performance persists up to a 24-hour lead time.

In the real-world case studies, we merged soil moisture data from in-suit monitoring sites with the near-real-time CLDAS soil moisture data. This fusion produces spatially distributed data characterized by high temporal immediacy while addressing the limitation of point-scale in in-suit soil data. Contrasting with experiments using synthetic data, extending soil moisture observation intervals to 8 hours impacts the performance of the $\text{AEnKF}_S$ scheme. In one-step prediction, the $\text{AEnKF}_{SQ}$ scheme exhibits the highest level of accuracy with the $\text{MR}_{RMSE}$ of 0.84. Concurrently, the simulation precision of the $\text{AEnKF}_Q$ scheme exceeds that observed in $\text{AEnKF}_S$, with $\text{MR}_{RMSE}$ values of 0.88 and 0.91, respectively. Variations in results are observed under different lead times. $\text{AEnKF}_{SQ}$ and $\text{AENKF}_S$ consistently demonstrate an assimilation effect duration of 8 hours, in contrast to the 5-hour temporal persistence of assimilation effect of $\text{AENKF}_Q$. The use of AEnKF for updating cumulative channel flow markedly enhances the accuracy of discharge forecasting in a brief lead time. In contrast, the adjustment extent of discharge by updating free water storage in a single-step forecast might be less than that achieved with $\text{AEnKF}_Q$. Nevertheless, it guarantees a more sustained assimilation effect. The $\text{AEnKF}_{SQ}$ integrates the strengths of the previous two

strategies, thereby improving discharge forecasting accuracy even when a particular strategy does not update effectively and prolonging the temporal persistence of the assimilation effect.

8. Are there any limitations or recommendations for the application of this study? Is the proposed methodology applicable to all regions? Can the method be applied in data-scarce regions with limited observations?

**Response**: Thank you for your comment. The Xin'anjiang model is based on the saturation-excess runoff generation mechanism, where net rainfall is first entirely used to replenish soil water, and once the soil moisture content in the unsaturated zone reaches field capacity, all subsequent net rainfall is used to generate runoff. This runoff generation mechanism is generally applicable to humid and semi-humid regions, making the Xin'anjiang model theoretically suitable only for these areas. Since humid regions in China are most affected by flood disasters and the Xin'anjiang model is currently the most widely used hydrological model in operational flood forecasting in China, this study uses the Xin'anjiang model as an example and tests it only in humid regions in China. However, it is important to emphasize that the AEnKF method with the enhanced error models proposed in this study can be easily coupled with any hydrological model, so its application is not limited to humid and semi-humid regions. In future research, we will focus on coupling and testing the AEnKF with enhanced error models and other hydrological models. The proposed method in this study involves assimilating observational data into the hydrological model, making it inapplicable in data-scarce regions. We have added a discussion of the limitations of this study in the revised manuscript (LINES 662-680).

**Revised Manuscript LINES 662-680:**
**6.3 Limitations**
The Xin'anjiang model is a conceptual hydrological model that generalizes the rainfall-runoff process. Its most prominent feature is performing runoff production calculations based on the saturation-excess runoff mechanism, meaning net rainfall is first entirely used to replenish soil water, and once the soil moisture content in the unsaturated zone reaches field capacity, all subsequent net rainfall is used to generate runoff. Therefore, the Xin'anjiang model is only suitable for humid and semi-humid regions where the saturation-excess runoff mechanism is dominant and is not applicable to arid and semi-arid regions. However, it is important to note that the state updating method proposed in this study is not limited to coupling with the Xin'anjiang model. In fact, this method can be easily coupled with any lumped or semi-distributed hydrological model that includes state variables related to soil moisture and channel storage. When coupled with

hydrological models suitable for semi-arid and arid regions, it can be effectively applied in those areas.

Semi-distributed hydrological models, like the Xin'anjiang model used in this study, have smaller state variable dimensions, allowing for the direct application of the proposed state updating scheme. However, in distributed models where each computational grid (e.g., DEM-based grids) has its own state variables, the state dimension becomes large, making direct application inefficient or prone to spurious correlations from distant observations. To resolve this, we recommend applying covariance localization to AEnKF (Janjić et al., 2011) or other localization techniques (Khaniya et al., 2022). For instance, in covariance localization, a localization radius (RL) is set, and the forecast error covariance matrix is adjusted using a correlation matrix derived from the Schur product theorem. This study focuses on jointly assimilating soil moisture and streamflow using AEnKF, and performing localization on AEnKF is beyond the scope of this research. We will explore this further in future work.

9. As a crucial step in the study, the statistical characteristics of the data used (e.g., peak discharge) should be presented in detail. The statistical properties, including skewness, coefficient of variation, confidence intervals, boxplots for outlier data, distribution characteristics, minimum, maximum, and median values, etc., should be provided in a table.

**Response**: Thank you for your suggestion. In Supplement (part S4), we have included a table (Table S4-2) with the statistical characteristics of the peak flow for the flood events in this study, including the mean, standard deviation, minimum, maximum, median, skewness, kurtosis, coefficient of variation, and 95% confidence interval.

**Revised Supplement Table S4-2:**

**Table S4-2. Statistical characterization of peak flow**

| | | | Mean ($m^3/s$) | standard deviation ($m^3/s$) | Minim-um ($m^3/s$) | Maxi-mum ($m^3/s$) | Median ($m^3/s$) | Skew-ness | Kurt-osis | Coefficient of Variation | 95% confidence interval ($m3/s$) |
|---|---|---|---|---|---|---|---|---|---|---|---|
| observed peak flow | | Calibr-ation | 15982 | 8049 | 7348 | 35725 | 14996 | 1.03 | 0.55 | 0.50 | (11534, 20431) |
| | | Valid-ation | 11255 | 6343 | 4747 | 25963 | 8420 | 0.89 | 0.37 | 0.56 | (7879, 14630) |
| Simulated Peak flow | | Calibr-ation | 15942 | 8142 | 7462 | 35648 | 14958 | 0.97 | 0.22 | 0.51 | (11444, 20440) |
| | | Valid-ation | 10596 | 5669 | 4244 | 23428 | 7850 | 0.87 | -0.15 | 0.53 | (7578, 13614) |

10. I recommend including the error estimation and evaluation metrics section from the supplement as an appendix in the main document, rather than providing it as a separate

file. This will facilitate a more cohesive understanding for interested readers.

**Response**: Thank you for your suggestion. We have carefully considered your suggestions and provided a summary of these two sections in the main text (LINES 239-273 and LINES 332-346), ensuring that general readers can gain a comprehensive understanding without being overwhelmed by excessive technical details. For data assimilation experts interested in the technical specifics, they can find detailed information in the supplementary materials. On the other hand, the main focus of this study is the joint assimilation of multi-source data using the improved AEnKF, and the main text is already quite lengthy in fully presenting this focus. After careful consideration, we believe that providing the technical details of error estimation and evaluation metrics as supplementary material is a better choice to improve the readability of the paper and manage the length of the main text.

**Revised Manuscript LINES 239-273:**

**2.3 Error estimation**

[revised manuscript text omitted]

---

## Author Response (AR2)

Dear Dr. Yi He,

I would like to express my sincere gratitude for the insightful and constructive feedback you provided on our manuscript. Your comments have been incredibly helpful in improving the clarity and quality of our work, and we truly appreciate the time and effort you dedicated to reviewing it. All the comments have been considered and a point-by-point response has been provided below.

The point-by-point response is formatted as follows:
   - Your comments are shown in blue
   - Our responses are shown in black
   - The changes in the manuscript are shown in red. The line numbers indicated in this response are those in the "Revised Manuscript with no changes marked" document
   - The unchanged parts of the manuscript are shown in black
* * *
1. In your revised manuscript, you state "the Xin'anjiang model is only suitable for humid and semi-humid regions where the saturation-excess runoff mechanism is dominant and is not applicable to arid and semiarid regions." It is not necessarily true that saturation-excess runoff mechanism is not present in some semiarid regions. This is still debatable, and it would be better to say "the Xin'anjiang model is mostly suitable for humid and semi-humid regions where the saturation-excess runoff mechanism is dominant and is less or not applicable to arid and semiarid regions."

**Response**: Thank you for your helpful suggestion. We have rewritten the sentence as per your advice to avoid any potential controversy (Lines 667-669).

**Revised Manuscript LINES 667-669:**

Therefore, the Xin'anjiang model is mostly suitable for humid and semi-humid regions where the saturation-excess runoff mechanism is dominant and is less or not applicable to arid and semiarid regions.

2. "As long as the warming-up period is adequately long", this is ambiguous. Please clarify what is an adequately long warming-up period.

**Response**: Thank you very much for your comment. Kim et al. (2018) suggested that different conceptual hydrological models require varying warming-up periods, which may be influenced by factors such as model structure. Their testing showed that when the warming-up periods for the two conceptual models, HYMOD and IHACRES, were

set to 1.5 months and 6 months respectively, the initial condition no longer had a significant impact on the results. In our study, the daily simulation began on February 10, 2014. Testing revealed that even under extreme conditions where the initial soil moisture is set to 0 or fully saturated, there is almost no impact on the flood simulation results (see Table R1). This indicates that by the time of flood onset, the state variables of the hydrological model had either stabilized. Since the duration of the warm-up period for testing is beyond the scope of our study, we only briefly mention the test and its results in the revised manuscript (Lines 371-376).

**Table R1. Flood simulation results for different daily simulation initial conditions**

| Serial number | Relative error of flood peaks | | | NSE | | |
|---|---|---|---|---|---|---|
| | Zero initial values | Saturated initial values | Half-saturated initial values | Zero initial values | Saturated initial values | Half-saturated initial values |
| 2014052300 | 0 | 0 | 0 | 0.85 | 0.86 | 0.86 |
| 2014070300 | -0.05 | -0.05 | -0.05 | 0.95 | 0.95 | 0.95 |
| 2014071400 | 0 | 0 | 0 | 0.94 | 0.94 | 0.94 |
| 2015060121 | -0.04 | -0.04 | -0.04 | 0.93 | 0.93 | 0.93 |
| 2015060718 | -0.08 | -0.08 | -0.08 | 0.83 | 0.83 | 0.83 |
| 2015062023 | -0.12 | -0.12 | -0.12 | 0.95 | 0.95 | 0.95 |
| 2016050703 | -0.07 | -0.07 | -0.07 | 0.91 | 0.91 | 0.91 |
| 2016062017 | -0.18 | -0.18 | -0.18 | 0.72 | 0.72 | 0.72 |
| 2016062720 | -0.13 | -0.13 | -0.13 | 0.87 | 0.87 | 0.87 |
| 2016070311 | -0.06 | -0.06 | -0.06 | 0.97 | 0.97 | 0.97 |
| 2017052208 | 0.01 | 0.01 | 0.01 | 0.89 | 0.89 | 0.89 |
| 2017062711 | 0 | 0 | 0 | 0.94 | 0.94 | 0.94 |
| 2017081121 | 0.14 | 0.14 | 0.14 | 0.66 | 0.66 | 0.66 |
| 2018053010 | 0.02 | 0.02 | 0.02 | 0.92 | 0.92 | 0.92 |
| 2018092518 | -0.12 | -0.12 | -0.12 | 0.83 | 0.83 | 0.83 |
| 2019051905 | -0.06 | -0.06 | -0.06 | 0.94 | 0.94 | 0.94 |
| 2019070700 | -0.05 | -0.05 | -0.05 | 0.96 | 0.96 | 0.96 |
| 2020070800 | -0.10 | -0.1 | -0.10 | 0.87 | 0.87 | 0.87 |
| 2020071823 | -0.17 | -0.17 | -0.17 | 0.86 | 0.86 | 0.86 |
| 2020091500 | -0.02 | -0.02 | -0.02 | 0.96 | 0.96 | 0.96 |
| 2021050300 | 0.05 | 0.05 | 0.05 | 0.54 | 0.54 | 0.54 |
| 2021051112 | -0.07 | -0.07 | -0.07 | 0.89 | 0.89 | 0.89 |
| 2021060300 | -0.08 | -0.08 | -0.08 | 0.82 | 0.82 | 0.82 |
| 2023040308 | 0.27 | 0.27 | 0.27 | 0.45 | 0.45 | 0.45 |
| 2023050416 | -0.11 | -0.11 | -0.11 | 0.64 | 0.64 | 0.64 |
| 2023052008 | 0.36 | 0.36 | 0.36 | 0.70 | 0.70 | 0.70 |
| 2023062100 | -0.16 | -0.16 | -0.16 | 0.86 | 0.86 | 0.86 |
| 2023063000 | -0.16 | -0.16 | -0.16 | 0.55 | 0.55 | 0.55 |
| 2023072516 | -0.10 | -0.10 | -0.10 | 0.87 | 0.87 | 0.87 |

| Serial number | Relative error of flood peaks | | | NSE | | |
|---|---|---|---|---|---|---|
| | Zero initial values | Saturated initial values | Half-saturated initial values | Zero initial values | Saturated initial values | Half-saturated initial values |
| 2024040100 | 0.16 | 0.16 | 0.16 | 0.47 | 0.47 | 0.47 |
| 2024042900 | 0 | 0 | 0 | 0.69 | 0.69 | 0.69 |

**Revised Manuscript LINES 371-376:**

As long as the warming-up period is adequately long, the influence of initial soil moisture on the daily simulation becomes minimal by the end of warming-up period, allowing soil moisture for daily simulation to be used as initial conditions for hourly simulation (Yao et al., 2012). The daily simulation in this study began on February 10, 2014. Testing showed that even in extreme cases where the initial soil moisture in the daily simulation is set to zero or fully saturated, there is almost no impact on the flood simulation results. So, they can be set arbitrarily within reason.

3. LT = 8 hours. Please comment on how useful it is in real world applications, for example, is it sufficient for issuing flood warning and mobilising rescue resources within 8 hours?

**Response**: Thank you for your helpful suggestion. The average time interval between the onset of the main rainfall event and the peak flow at the basin outlet is approximately 7 hours in study catchment. Therefore, we set the maximum lead time to 8 hours, slightly longer than this interval. This choice is supported by several reasons:

(1) This lead time meets the requirements for real-time flood forecasting in medium-sized basins. Real-time flood forecasting is a dynamic prediction system based on real-time monitoring data, combined with hydrological and/or hydrodynamic models, to forecast the development of flood events. It provides essential information such as the timing of peak flows, water levels, and discharge when flooding occurs. Due to its high timeliness and short forecasting window, the lead time is generally set to a few hours (e.g., Toth et al., 2000; Liu et al., 2016). The method proposed in this study is particularly suited for real-time updating in real-time flood forecasting, where it dynamically updates the hydrological model's state variables based on real-time observational data. Our approach can provide reliable real-time and near real-time information for emergency responses, assisting governments and relevant flood control agencies in managing evacuations, resource allocation, reservoir operations, and other disaster mitigation efforts, thereby minimizing casualties and property losses caused by floods.

(2) Some researchers have identified that the accumulation of errors in the state variables of hydrological models is the main source of uncertainty during the initial phase and near real-time forecasting period (Weerts et al., 2006). However, as the lead time extends, the uncertainty in numerical weather predictions increasingly dominates and becomes the primary factor affecting flood forecasting accuracy (Yossef et al., 2013; Thiboult et al., 2016). The aim of this study is to reduce the accumulated error in hydrological model state variables at the start of the forecast. By setting a relatively short lead time, the error accumulation in state variables remains the key factor influencing flood forecast accuracy. In medium- and long-term flood forecasting, however, the uncertainties from numerical weather predictions may take precedence, which lies beyond the scope of our study.

We have added a discussion on the lead time in the discussion section of the revised manuscript (Lines 682-697).

**Revised Manuscript LINES 682-697:**

Real-time flood forecasting is a dynamic prediction system based on real-time monitoring data, combined with hydrological and/or hydrodynamic models, to predict the evolution of flood processes. It provides critical information such as the time of peak flow, water levels, and discharge when a flood occurs. This type of forecasting is characterized by its high timeliness and short forecasting window, with the lead time generally set to several hours (e.g., Toth et al., 2000; Liu et al., 2016). The methods proposed in this study is particularly suited for state updating within real-time flood forecasting, as it dynamically updates the state variables of the hydrological model using real-time observational data, reducing the accumulation of errors. In real-world cases, we set the maximum lead time to 8 hours, which sufficiently meets the requirements for real-time flood forecasting in medium-sized catchments. This provides reliable real-time and near-real-time information for emergency responses, assisting government and flood control agencies in organizing evacuations, resource allocation, and reservoir operations, thereby minimizing casualties and property damage caused by floods. Moreover, to test the temporal persistence of the state updating method, we used historical observed rainfall as a perfect proxy for numerical weather forecasts, thereby avoiding the introduction of uncertainties from numerical weather predictions. As the lead time increases, uncertainties in numerical weather predictions may gradually replace the accumulation of errors in hydrological model state variables as the primary source of uncertainty in flood forecasting (Weerts et al., 2006; Yossef et al., 2013; Thiboult et al., 2016). For medium- to long-term flood

forecasts, greater attention may need to be given to uncertainties stemming from numerical weather predictions.

**Reference mentioned in our responses**

Kim, K. B., Kwon, H. H., and Han, D. W.: Exploration of warm-up period in conceptual hydrological modelling, J. Hydrol., 556, 194-210, https://doi.org/10.1016/j.jhydrol.2017.11.015, 2018.

Liu, Z., Guo, S., Zhang, H., Liu, D., and Yang, G.: Comparative study of three updating procedures for real-time flood forecasting, Water Resour. Manag., 30, 2111-2126. https://doi.org/10.1007/s11269-016-1275-0, 2016.

Thiboult, A., Anctil, F., and Boucher, M. A.: Accounting for three sources of uncertainty in ensemble hydrological forecasting, Hydrol. Earth Syst. Sci., 20, 1809-1825, https://doi.org/10.5194/hess-20-1809-2016, 2016.

Toth, E., Brath, A., and Montanari, A.: Comparison of short-term rainfall prediction models for real-time flood forecasting, J. Hydrol., 239(1-4), 132-147. https://doi.org/10.1016/S0022-1694(00)00344-9, 2000.

Weerts, A. H. and El Serafy, G. Y. H.: Particle filtering and ensemble Kalman filtering for state updating with hydrological conceptual rainfall-runoff models, Water Resour. Res., 42, W09403, https://doi.org/10.1029/2005WR004093, 2006.

Yossef, N. C., Winsemius, H., Weerts, A. H., van Beek, R., and Bierkens, M. F. P.: Skill of a global seasonal streamflow forecasting system, relative roles of initial conditions and meteorological forcing, Water Resour. Res., 49, 4687-4699, https://doi.org/10.1002/wrcr.20350, 2013.

---

## Author Response (AR3)

Dear Dr. Yi He,

I would like to express my sincere gratitude for the time and effort you dedicated to reviewing our manuscript. Your specific recommendations have been incredibly helpful in improving the quality of our work, and we truly appreciate your support. All the comments have been considered and a point-by-point response has been provided below.

The point-by-point response is formatted as follows:
   - Your comments are shown in blue
   - Our responses are shown in black
   - The changes in the manuscript are shown in red. The line numbers indicated in this response are those in the "Revised Manuscript with no changes marked" document
   - The unchanged parts of the manuscript are shown in black
* * *
1. Line 164: "2 Methodology and method": change it to "2 Methodology"

**Response**: Thank you for your helpful suggestion. We have changed the section title as per your advice (Lines 164).

**Revised Manuscript LINES 164:**

2 Methodology

2. Line 166-170: "The Xin'anjiang model, conceptualized by Zhao (1992), is a distinguished hydrological model, primarily based on a saturation excess mechanism. Renowned for its straightforward structure and explicit parameter definitions, this model excels in simulating humid catchments, making it a popular tool for flood forecasting in in China. To account for spatial variability in rainfall distribution and surface characteristics, the model typically segments a catchment into several sub-basins. These sub-basins act as computational units for runoff generation and routing."

"distinguished hydrological model": Inappropriate in academic writing. If you want to highlight a model's respected status, you could describe it as "widely recognised," "commonly used," or "well-regarded in hydrological research." This phrasing is more objective, in line with academic standards.

"To account for spatial variability in rainfall distribution and surface characteristics, the model typically segments a catchment into several sub-basins.": change to "To account for spatial variability, the model typically divides a catchment into sub-catchments."

Please also change sub-basins to sub-catchments throughout the manuscript.

**Response**: Thank you for your helpful suggestion. We have revised the wording of the relevant sentences according to your suggestions (Lines 166-170). In addition, we have replaced sub-basin with sub-catchment and basin with catchment throughout the text to maintain consistency in terminology.

**Revised Manuscript LINES 166-170:**

The Xin'anjiang model, conceptualized by Zhao (1992), is a commonly used hydrological model, primarily based on a saturation excess mechanism. Renowned for its straightforward structure and explicit parameter definitions, this model excels in simulating humid catchments, making it a popular tool for flood forecasting in in China. To account for spatial variability, the model typically divides a catchment into sub-catchments. These sub-catchments act as computational units for runoff generation and routing.

3. Line 171: The Xin'anjiang model demands relatively straightforward driving data

Change to "The Xin'anjiang model demands relatively simple forcing data"

**Response**: Thank you for your helpful suggestion. We have revised this sentence according to your advice (Lines 171-172).

**Revised Manuscript LINES 171-172:**

The Xin'anjiang model demands relatively simple forcing data, and key inputs include the areal mean rainfall depth (P) and pan evaporation (EM) for each sub-catchment.

4. Line 348: "Study areas and data"

Change to "Study area and data"

**Response**: Thank you for your helpful suggestion. We have changed the section title as per your advice (Lines 348).

**Revised Manuscript LINES 348:**

3 Study area and data

5. Line 350: "with elevations ranging from 42 to 1,396 meters"

Meters above sea level? Need to specify the reference.

**Response**: Thank you for your helpful suggestion. We have added an explanation in the text that the elevation is referenced against sea level (Lines 350-351).

**Revised Manuscript LINES 350-351:**

It covers an area of approximately 8,033 km², with elevations ranging from 42 to 1,396 meters above sea level.

6. Line 372-373: please provide the recommended warm-up length

**Response**: Thank you for your helpful suggestion. We recommend a warm-up period of at least 3 months, and have added to the manuscript (Lines 371-377).

**Revised Manuscript LINES 371-377:**

Consequently, a daily simulation must be performed prior the hourly simulation, and we recommend that this warming-up (spin-up) be at least three months long. This period enables the soil moisture simulated daily, driven by observed hydrometeorological data, to gradually approaches actual soil moisture (Kim et al., 2018). The influence of initial soil moisture on the daily simulation becomes minimal by the end of

warming-up period, allowing soil moisture for daily simulation to be used as initial conditions for hourly simulation (Yao et al., 2012). The daily simulation in this study began on February 10, 2014. Testing showed that even in extreme cases where the initial soil moisture in the daily simulation is set to zero or fully saturated, there is almost no impact on the flood simulation results.

7. Lines 461, 487: two sub-headings should be removed. It is clear enough to tell from the text below what the following paragraph is about. The same applies to the sub-headings used in Section 5.2.2.
**Response**: Thank you for your helpful suggestion. We have removed the subheadings under Sections 5.1.3 and 5.2.2.

8. Figures 11, 12: fonts in the legends need to be enlarged. They cannot be read clearly at the 100% scale.
**Response**: Thank you for your helpful suggestion. We have increased the font size of the legends in Figures 11 to 13. Additionally, we have optimized the layout of Figures 12 and 13 to improve readability.
**Revised Manuscript Figure 11:**

[Figure]

Fig. 11. The accuracy of forecasted discharge under different lead time. (a) NNSE, (b) $R_{RMSE}$.

[Figure]

**Fig. 12. Hydrograph during flood event labeled No.2023040308. (a-b) AEnKF$_Q$ Scheme, (c-d) AEnKF$_S$ Scheme, (e-f) AEnKF$_{SQ}$ Scheme. The left panel shows the discharge at the catchment outlet, and the right panel displays the free water storage in sub-catchment 1.**

[Figure]

**Fig. 13. Hydrograph during flood event labeled No.2023052008. (a-b) AEnKF$_Q$ Scheme, (c-d) AEnKF$_S$ Scheme, (e-f) AEnKF$_{SQ}$ Scheme. The left panel shows the discharge at the catchment outlet, and the right panel displays the free water storage in sub-catchment 1.**

---

## Author Response (AR4)

Dear Editor,

In the Acknowledgements section of the final version of the manuscript, we have added appreciation for the financial support from the China Postdoctoral Science Foundation (Grant No. 2024M760743), which will cover the article processing charges.